

# Modelling and verification of post-quantum key encapsulation mechanisms using Maude

Víctor García[1], Santiago Escobar[1], Kazuhiro Ogata[2], Sedat Akleylek[3,4] and Ayoub Otmani[5]

[1] Universidad Politécnica de Valencia, Valencia, Spain
[2] Japan Advanced Institute of Science and Technology, Ishikawa, Japan
[3] Ondokuz Mayis University, Samsun, Turkey
[4] University of Tartu, Tartu, Estonia
[5] University of Rouen Normandie, Rouen, France

## ABSTRACT

Communication and information technologies shape the world's systems of today, and those systems shape our society. The security of those systems relies on mathematical problems that are hard to solve for classical computers, that is, the available current computers. Recent advances in quantum computing threaten the security of our systems and the communications we use. In order to face this threat, multiple solutions and protocols have been proposed in the Post-Quantum Cryptography project carried on by the National Institute of Standards and Technologies. The presented work focuses on defining a formal framework in Maude for the security analysis of different post-quantum key encapsulation mechanisms under assumptions given under the Dolev-Yao model. Through the use of our framework, we construct a symbolic model to represent the behaviour of each of the participants of the protocol in a network. We then conduct reachability analysis and find a man-in-the-middle attack in each of them and a design vulnerability in Bit Flipping Key Encapsulation. For both cases, we provide some insights on possible solutions. Then, we use the Maude Linear Temporal Logic model checker to extend the analysis of the symbolic system regarding security, liveness and fairness properties. Liveness and fairness properties hold while the security property does not due to the man-in-the-middle attack and the design vulnerability in Bit Flipping Key Encapsulation.

# INTRODUCTION

Today's security is heavily based on computationally hard problems. Most of the current network infrastructure and systems work over classical computers. Specifically, most of these protocols rely on three problems considered hard to solve under classic computation: the integer factorization problem, the discrete logarithm problem and the elliptic-curve

Corresponding author
Víctor García, vicgarval@upv.es

discrete logarithm problem. Such problems are considered to be in the NP category, which stands for non-deterministic polynomial time for classic computers.

Research in the quantum field has been active in the past years, proposing new algorithms and methods that could endanger the security of current crypto-systems and cryptographic schemes. As stated before, the protocols of today are based on mathematical problems that are hard to solve for classical computers, but such problems become solvable with quantum computers. Some of the most popular asymmetric (or public key) algorithms, which rely on integer factorization, will become insecure under quantum computers using Shor's algorithm from *Shor (1994)*. Another example is Grover's search algorithm, proposed by *Grover (1996)*, which makes it possible to reduce the complexity of the integer factorization problem to a quadratic cost. A current simple solution to counter the reduction cost is to extend the length of the key.

In order to face the threat quantum computers suppose to the security of most information systems, the National Institute for Standards and Technologies (NIST) started 2017 the Post-Quantum Cryptography Project (PQC). The project is conducted as a contest, divided into multiple rounds, to analyze candidate protocols and select some as a standardized solution to face the threat of quantum adversaries. As of the submission of this article, there have been four rounds of the project, and candidates range between public-key encryption and key establishment to digital signature algorithms.

We focus on key encapsulation mechanisms (KEMs). A KEMs primary goal is to securely share a key between two network participants where channels are not safe from intruders. The goal is interesting for conventional cryptography, also known as symmetric cryptosystem, which uses a secret key to encrypt a message. The selected protocols are Kyber, BIKE, and Classic McEliece. The former, Kyber, is lattice-based, meaning its security is based on the hardness of solving the Learning With Errors (LWE) problem over module lattices. Kyber was selected as the finalist in round 3. The latter two are code-based. Specifically, BIKE bases its security on Quasi-Cyclic Moderate Density Parity Check (QC-MDPC) codes. Meanwhile, McEliece bases its security on error-correcting codes and the difficult problem of decoding a message with random errors. For some KEMs, some vulnerabilities have been found such as man-in-the-middle (*Tran et al., 2022a*; *Tran et al., 2022b*; *Tran et al., 2022c*; *García, Escobar & Ogata, 2022*).

Once the problem of quantum computers and some possible solution schemes are chosen by NIST, we need to establish how to analyze and reason about the protocols. For the analysis of security systems and protocols, two main approaches can be taken: computational security and symbolic security; see *Blanchet (2012)* for further information on both approaches and their comparison. The former is based on mathematical proofs over a computational model, where messages are bit strings, and the adversary is any probabilistic Turing machine. Cryptographers generally use computational security, and the authors of the three selected KEM have already covered this approach. Although the computational model is closer to reality, it complicates the proofs and is hard to understand for non-experts of cryptography. The latter is based on symbols, where the cryptography primitives are function symbols acting as black boxes. It is important to note that these models assume perfect cryptography (*Dolev & Yao, 1983*), *i.e.*, ciphertexts cannot be broken

without the proper key. Furthermore, symbolic models are suitable for automation and easier to understand. It is essential to mention that this approach not only can be applied to the selected protocols but also to any other scheme or mechanism in rounds 3 or 4 of the Post-Quantum Cryptography project, either public key encryption and key establishment algorithms or digital signatures.

To further extend the implications of performing symbolic security analysis, we could have a protocol under inspection. First, it is necessary to write a formal system specification of the protocol in a formal specification language, such as Maude. Because Maude is a programming language and a logical framework we have not only a formal model of the protocol to reason about but an effective implementation. This modelling process implies the necessity to formalize every data structure used in the protocol and every possible action conducted by the participants of the protocol. Furthermore, if the protocol is cryptographic then it is necessary to include the presence of dishonest participants and consider all possible measures that can be utilized by the attackers, such as extra algebraic properties, probably not for implementation but certainly for security analysis. The formal specification and analysis process is likely to find out subtle problems in the protocol that otherwise could go unnoticed; see *Barbosa et al. (2021)* for a recent SoK on symbolic protocol analysis. To do formal verification, it is necessary to formalize desired properties of the protocol in a logic, such as linear temporal logic (LTL). Furthermore, model checking properties over the protocol model may reveal extra problems to those detected during the formal specification.

*Contributions.* We perform symbolic security specification and analysis using Maude and LTL on three post-quantum key encapsulation mechanisms (KEMs). Our contributions are (i) a novel framework for the specification of post-quantum cryptographic protocols using Maude, (ii) an extended analysis of three post-quantum KEM protocols using model checking, (iii) verification of the presence of a man-in-the-middle attack in the three selected KEMs and (iv) discovery of a design flaw in BIKE, allowing a malicious participant to use a weak key to impersonate another participant. All developed modules resulting from this work can be found at https://github.com/v1ct0r-byte/PQC-in-Maude.

*Outline.* The rest of the article is structured as follows. 'Related Work' presents an overview of the advances made in the symbolic analysis of protocols, mentioning tools and articles of similar nature. 'Rewriting Logic and Maude' introduces rewriting-logic and gives a presentation on the fundamental building blocks of Maude. 'Key Encapsulation Mechanisms' explains the behaviour and security principles of the three selected KEMs. 'Framework Specification' describes the core modules of our framework to build symbolic models of these KEMs. 'Key Encapsulation Mechanism Specifications' dives into the concrete specification of each KEM through the use of our framework. 'Verification' dives into the two analysis approaches carried on over the symbolic model to formally analyze the constructed KEMs. Finally, 'Conclusion' summarizes the article and gives future directions for the presented work. This article is an extended version of *García, Escobar & Ogata (2022)* and *García (2022)*.

## RELATED WORK

Currently, advances in protocol security analysis have been made. One interesting idea is the one proposed at *Blanchet (2012)*, where the author explains several examples of formal specification of protocols and introduces and explains the symbolic and computational model analysis. In *Cortier, Kremer & Warinschi (2011)*, the authors explore the current literature and articles on both symbolic and computational analysis of protocols. In this survey, they analyze the results by combining both types of analysis. This proposal was made initially by *Abadi & Rogaway (2002)* in order to close the gap between both lines of protocol verification.

Among the various symbolic protocol analysis tools available, we have Maude-NPA (*Escobar, Meadows & Meseguer, 2009*; *Escobar, Meadows & Meseguer, 2006*), related to the programming language Maude (*Clavel et al., 2007*; *Durán et al., 2020*). Maude-NPA has a theoretical basis on rewriting logic, unification and narrowing and performs a backwards search from a final attack state to determine whether or not it is reachable from an initial state. Some symbolic tools, such as ProVerif (*Blanchet et al., 2018*; *Blanchet, Cheval & Cortier, 2022*), are based on an abstract representation of a protocol using Horn clauses. The verification of security properties is done by reasoning on these representative clauses. Other symbolic tools, such as Tamarin (*Meier et al., 2013*; *Basin et al., 2022*), are based on constraint solving to perform an exhaustive, symbolic search for executions traces. Furthermore, other symbolic tools such as Scyther (*Cremers, 2008*) or CPSA (*Ramsdell & Guttman, 2018*) attempt to enumerate all the essential parts of the different possible executions of a protocol. Also, AKISS (*Gazeau & Kremer, 2017*) or the DEEPSEC prover (*Cheval, Kremer & Rakotonirina, 2018*) are other tools mostly used to decide equivalence properties.

Some related work can be found for the symbolic security analysis of protocols with quantum features. For example, the authors of *Gazdag et al. (2021)* built a model of IKEv2 on a classical setting and then perform some analysis on it for seven properties, all of it using the Tamarin prover. With this first symbolic analysis, they prove their model to be correct and corroborate previous results by other authors. Later, they extended the model with the improvements included in the latest extension of IKEv2 in order to include a quantum-resistant key exchange. With this extension, they perform a new analysis, where all properties hold, verifying the security of the new extension.

Another article performing symbolic analysis of post-quantum protocols is *Jacomme et al. (2023)*. The authors use the recently published tool SAPIC$^+$ to analyze the Ephemeral Diffie Hellman Over COSE (EDHOC) protocol, on its 12th draft version. With SAPIC$^+$ they can automatically transform their model to a suitable one in Tamarin, ProVerif or DEEPSEC respective syntax. This allows the authors to take advantage of the strength of each tool to perform analysis over their modular composed model of the protocol. With the analysis, they discover several flaws and report them to the team of EDHOC. Then propose some fixes, validate them and are accepted into the 14th draft version of the protocol.

In *Hülsing et al. (2021)* a variant for a handshake protocol from the WireGuard VPN protocol with post-quantum capabilities is presented. They performed such adaptation

by replacing the previous Diffie-Hellman-based handshake with key-encapsulation mechanisms. The authors verify the security of their proposal with symbolic and computational proofs. On the one hand, the symbolic proofs verify more security properties than the computational proofs and are computer verified. On the other hand, computational proofs give stronger security guarantees as the proof makes less idealizing assumptions.

In *Tran et al. (2022a)*, a first approximation on the symbolic specification of post-quantum protocols in Maude-NPA is provided. The authors selected to specify the Post-Quantum TLS protocol primitives and execution trace, leaving for future work the definition of attack states and the verification of the protocol. This type of work is interesting and necessary to us because it demonstrates the capability of Maude-NPA as well as Maude to verify more advanced schemes automatically.

The closest to our work are *Tran et al. (2022b)*; *Tran et al. (2022c)*, which present the first symbolic security analyses of a collection of post-quantum protocols, among which Kyber can be found. The authors report a man-in-the-middle (MITM) attack for each of the specified protocols using the Maude search command. This analysis has paved the way for us by performing symbolic specification and analysis using Maude. However, our article differs in many ways: (i) the syntax of the protocol and intruder models are completely different, (ii) we provide a general framework and show how it can be parameterized for the three selected KEMs, (iii) we consider code-base KEMs, specifically two, apart from lattice-based KEMs, (iv) we consider simpler and more general cryptographic properties, (v) we perform LTL model checking, and (vi) we have discovered a design flaw in the BIKE protocol. We only share the discovery of the man-in-the-middle attacks and similar honest and intruder actions.

## REWRITING LOGIC AND MAUDE

Maude (*Clavel et al., 2007*; *Durán et al., 2020*) is based on rewriting logic (*Meseguer, 1992*), a logic ideally suited to specify and execute computational systems in a simple and natural way. Since nowadays most computational systems are concurrent, rewriting logic is particularly well suited to specify concurrent systems without making any early commitments about the model of concurrency in question, which can be synchronous or asynchronous, and can vary widely in shape and nature: from a Petri net (*Stehr, Meseguer & Ölveczky, 2001*) to a process calculus (*Martí-Oliet & Verdejo-López, 2000*), from an object-based system (*Meseguer, 1993*) to asynchronous hardware (*Katelman, Keller & Meseguer, 2012*), from a mobile *ad hoc* network protocol (*Liu, Ölveczky & Meseguer, 2015*) to a cloud-based storage system (*Bobba et al., 2018*), from a web browser (*Chen et al., 2007*) to a programming language with threads (*Meseguer & Roşu, 2007*), or from a distributed control system (*Bae, Meseguer & Ölveczky, 2014*) to a model of mammalian cell pathways (*Eker et al., 2001*; *Talcott et al., 2003*). And all *without any encoding*: You see and get a direct system definition without any artificial encoding.

Rewriting logic has a sub-logic called membership equational logic. This sub-logic defines a system's deterministic parts using functional modules. In contrast, Maude

system modules represent concurrent systems as conditional rewrite theories that model a nondeterministic system which may never terminate and where the notion of a computed value may be meaningless. In this concurrent system, the membership equational sub-theory defines the states of such a system as the elements of an algebraic data type, such as terms in an equivalence class associated with cryptography properties. We can call this aspect the static part of the specification. Instead, its dynamics, *i.e.,* how states evolve, are described by the transition rules, which specify the possible local concurrent transitions of the system thus specified. The system's concurrency is naturally modelled by several transition rules in a given state that may be applied concurrently to different sub-parts, producing several concurrent local state changes. Thus rewriting logic models those concurrent transitions as logical deductions (*Meseguer, 1992*).

The most basic form of system analysis, in the form of explicit-state model checking, is illustrated by the use of the search command in Maude that performs reachability analysis from an initial state to a target state. Reachability can be used to verify invariants or find violations of invariants in the following sense. We can search for a violation of an invariant. If the invariant fails to hold, it will do so for some finite sequence of transitions from the initial state, which will be uncovered by the search command above since all reachable states are explored in a breadth-first manner. If the invariant does hold, we may be lucky and have a finite state system, in which case the search command will report failure to find a violation of the invariant. However, the search will never terminate if an infinite number of states are reachable from the initial state.

Under the assumption that the set of states reachable from an initial state is finite, Maude also supports explicit-state model checking verification of any properties in linear-time temporal logic (LTL) through its LTL model checker.

In the following sections, we will review different aspects of Maude and see an example specifying and analysing the Needham–Schroeder public key (NSPK) protocol. Each section introduces the notation in Maude and an example of how to use that notation. We selected the NSPK protocol because it is easy to understand. Figure 1 depicts a simplified version of the NSPK protocol without the intervention of a server to provide the public keys. The protocol's basis is that two participants, A and B, want to share a key. To do so, A sends a nonce, $N_a$, concatenated to his identifier and encrypted using B's public key. Upon receiving the message, B extracts $A; N_a$ since he uses his paired secret key, and thanks to knowing A's identifier, he obtains $N_a$. Now, B concatenates a nonce of his own, $N_b$, with the received one, encrypting both in a new message sent to A using the corresponding public key. Once A receives the messages, since A knows $N_a$, he can extract $N_b$ and send it in a third and last message to B in order to confirm that he can decipher it. In the end, A has in its possession $N_b$, and B has $N_a$, thus succeeding in sharing secrets.

## Sorts and subsorts

At Maude, the first thing we do when creating a module, whether it is functional or a system module, is to define the types used in the operators and variables we want to use. We must use the reserved word sort followed by an identifier to define a type. An identifier

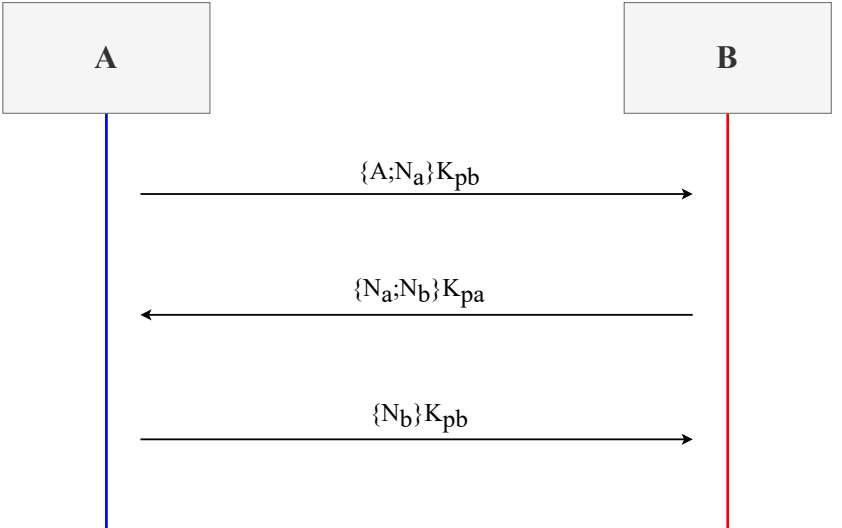

**Figure 1** Network diagram of a simplified Needham–Schroeder protocol version.

is a string of unreserved characters that Maude uses to identify everything from a module name to a variable name, or in this case, a type.

Let us see an example of how to declare sorts in order to introduce the concept of sub-type. The statement "sort Key ." defines a type with the identifier Key. The dot in Maude is used to terminate statements, similar to semicolons in Java, Python or C. If we want to define several types simultaneously, we can use the sorts keyword. Thus, the statement "sorts PubKey SecKey ." declares the types PubKey and SecKey.

Then we have the concept of *subsort*, a relationship between different types. To clarify this concept, we will use the previously defined types, *i.e.,* Key, PubKey and SecKey. To declare a relationship of sub-types, there is in Maude the reserved word subsort. Therefore, if we want to declare that the type *PubKey* is a sub-type of *Key*, we write the statement "subsort PubKey < Key ." where the symbol < is the operator reserved for indicating the direction of the relationship. We can do the same for SecKey with "subsort SecKey < Key .".

With the previous statements, we obtain a relationship resembling other languages' inheritance. As we see in Fig. 2, we have the three previous types in a tree-like structure, semantically representing that public and secret keys are considered keys.

## Operators

The operators allow us to define the syntax for the types we have defined, which allows us to define a notation that will express whatever we need. In this way, we can model the elements of a system or function with great freedom of expression.

Operators in Maude are defined by the reserved word op followed by the notation definition. Operators can be parameterized using the underscore character to indicate the position of each of the parameters within the notation. We can also parameterize them if

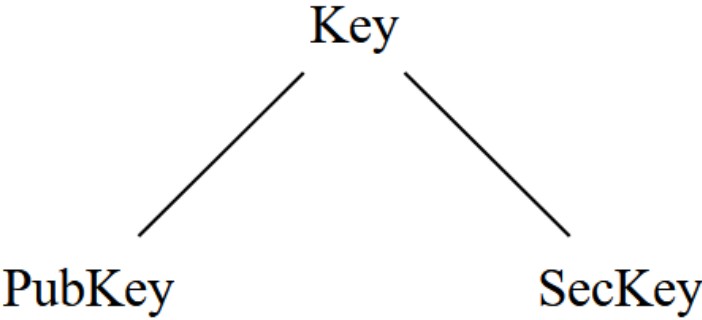

**Figure 2** **Subtype relation between defined Key, PubKey and SecKey types.**

```
op p[_](_) : Identifier Content -> Participant [ctor] .

op msg(_->_,_) : Identifier Identifier Key -> Message [ctor] .
```

**Figure 3** **Definition of operators for participants and messages in our model of the simplified NSPK.**

we omit underscore characters, making it a prefix operator; we will pass the parameters using parentheses as if it were a Java or C function. Once the parameters are set, we will indicate, after a colon, the types of each parameter in the order of appearance and, finally, the type returned by the operator after an arrow.

If we continue with the previous example of the simplified NSPK, we can define an operator to represent participants of the protocol and another one for the messages that they would exchange between them. We use Maude and get the following code snippet in Fig. 3 to show how participants and messages are defined. Notice that new *sorts* are used, which were previously defined in the same way we did for the examples of types of keys. Among these, we find `Identifier` to represent the values that act as identifiers of participants. Also, `Content` represents any information such as identifiers, keys or nonces defined by the protocol.

The first line of Fig. 3 shows that a participant is defined as an operator receiving two parameters. The participant's identifier is the first parameter, which will go between square brackets. The second parameter is all the content the participant has in his possession, which is enclosed between round brackets. As for the second line in Fig. 3, messages are defined as an operator starting with `msg` and then two identifiers separated by an arrow. This represents the origin and destination of a message. The third parameter is the encapsulated key inside the message, separated from the pseudo-header by a comma.

## Axioms
Axioms are properties or attributes on sets that are satisfied on a certain operator when defined in another operator's sequence. These attributes are applied in Maude if we write them between the symbols [ and ] at the end of the sequence operator, that is, the operator

```
sort Participants .
subsort Participant < Participants .
op empty : -> Participant .
op __ : Participant Participant -> Participants
        [assoc comm id: empty] .
```

**Figure 4** Definition in Maude of a set operator for participants with associative and commutative properties and `empty` as the identity element.

that contains a sequence of other operators of the same type. The default axioms defined by Maude are:

- `assoc` for specifying associativity property on the set
- `comm` for specifying commutativity property on the set
- `id:e` for specifying the identity element `e` for the set

To see an example of how to apply attributes, we will use the `Participant` type to define an operator that indicates a set of participants on which we will apply properties of interest. We first create the type `Participants` as shown in the first line of Fig. 4. The next line indicates, through a relation of sub-types, that a participant is a type of set of participants, that is, a set where there is only one element. Then, we define a participant as the identity element of a set of participants, the empty set. Finally, with the last line, we define the set of participants as a sequence of at least two participants, being this set associative, commutative and with `empty` as the identity element.

### Variables

Variables in Maude are defined using the structure `"var <id> : <var-sort>."`, where `<id>` is the name by which to identify the variable and `<var-sort>` the type it can contain. If we want to define more than one variable of the same type in a single statement, we can replace the previous `var` with `vars`.

As an example, we can define three variables with identifiers K, PK and SK. The identifier K is for a variable of type `Key`. For PK, the type will be `PubKey`, while for SK, it will be `SecKey`. An operator can be stored in each variable that returns the type specified for the variable. For example, a public key can be stored only in variables of type `PubKey` or super-types, like `Key`. Furthermore, thanks to the sub-type relationship, the variable K can also store the contents of PK or SK.

### System modules

System modules allow us to model systems in Maude, having non-deterministic behaviour with the possibility of infinite executions. We define them using the structure mod `<ModuleName> is <DeclarationsAndStatements> endm`, where `<ModuleName>` is the module identifier in uppercase, and `<DeclarationsAndStatements>` are statements. As statements, we can use those we have already seen, such as types and operators, although there are also equations and rules.

```
sort GlobalState .
op _|_ : Participants Messages -> GlobalState .
```

**Figure 5**  Definition of the system state for the NSPK protocol.

System modules specify rewrite theories. Theories are described in the form R = ($\sum$, $E$, $\varphi$, $R$). The element $\sum$ represents the set of all system states. The symbol $E$ represents the equational theories defined by the equations of the module. The symbol $\varphi$ is a function that assigns a set of natural numbers to the states of $\sum$ according to their number of arguments. The last element, $R$, is the set of system rules that specify the transitions between the states in $\sum$.

On the one hand, equations are statements of the form "eq <Term-1> = <Term-2>.", and they transform the left part of the equal sign to the right part so that Term-1 becomes Term-2. One way to view equations is as deterministic transformations that allow us to reduce or simplify expressions. On the other hand, rules are like equations, except the system's state changes within the modelled system upon applying a rule. Rules have the form "rl [id]: l => r .", having an optional identifier id and converting the left part l to the right part r.

These rules work over a state definition of our symbolic system. The specific representation of that system state is given in Fig. 5. Here, a sort is defined to represent the type of global state, and then we define an operator to model such type. The global state is defined as two components divided by |. On the left is a pool of participants in the system, while on the right, a pool of messages models the state of the system's network.

Now that we have the representation of the system, we move to the definition of the behaviour through the use of rules. The rules are presented in two groups. The first group has to do with the honest participant's capabilities. Figure 6 shows the six rules, three of them conditional, that symbolically represent how the system works regarding the NSPK protocol specification. These rules are labelled in a way that they come in pairs. Notice that the first rule, labelled as 1s, models the behaviour of constructing and sending the first message depicted in Fig. 1. As per the second rule, labelled 1r, it models the reception and treatment of the first message.

The rules modelling an intruder's behaviour are defined in Fig. 7. Here the capabilities of some intruder, say E, to supplant the identity of an honest participant is shown. The first rule, labelled as E1, states the ability of an intruder to modify a valid message sent to him in order to resend it to another participant. The second rule, E2, is just a redirection of the message. Then, the third and fourth rules model how the honest participant, that initiated the process with an intruder, processes the received message. Finally, the fifth rule processes the message to get the secret and creates a new one to trick the victim into thinking he could process it.

In relation to the rules, it is worth mentioning the capabilities of conditional rules. Conditional rules are similar to normal rules except they have the syntax "crl [<label>]: <Term-1> => <Term-2> if <Condition-1> /\... /\ <Condition- k>.", where

```
crl [1s] : p[A](pk(I,PK),Na,CSA) p[I](CSB) PS | none
              =>
           p[A](pk(I,PK),Na,CSA) p[I](CSB) PS | msg(A -> I, {Na,A}PK)
           if not Na in CSB .

rl [1r]  : p[B](sk(B,SK),pk(B,PK),CSB) PS | msg(I -> B, {N,I}PK)
              =>
           p[B](N,sk(B,SK),pk(B,PK),CSB) PS | none .

crl [2s] : p[B](Na,Nb,pk(A,PK),CSB) p[A](CSA) PS | none
              =>
           p[B](Na,Nb,CSB) p[A](CSA) PS | msg(B -> A, {Na,Nb}PK)
           if not Nb in CSA .

rl [2r]  : p[A](sk(A,SK),pk(A,PK),N,CSA) PS | msg(B -> A, {N,C}PK)
              =>
           p[A](C,N,CSA) PS | none .

rl [3s]  : p[A](Na,Nb,pk(B,PK),CSA) PS | none
              =>
           p[A](Na,Nb) PS | msg(A -> B, {Nb}PK) .

rl [3r]  : p[B](N,Na,pk(B,PK),sk(B,SK),CSB) PS | msg(A -> B, {N}PK)
              =>
           p[B](N,Na) PS | none .
```

**Figure 6** **Rules regarding the capabilities of the honest participants as defined by the simplified version of the NSPK protocol.**

the list of statements `<Condition-1>` `/\... /\` `<Condition-k>` are expressions that will return false or true. Conditions in conditional rules can take three forms:

1. Equations with the syntax `t=t'`.
2. Matching equations with the syntax `t:=t'`.
3. Rewrite expressions with the syntax `t->t'`.

Once our system is defined, we can analyze it to find vulnerabilities or failures due to the design. One form of analysis, reachability analysis, can be done using the `search` command. This command allows us to explore paths from one fully defined state to another that matches the given pattern. This command allows using different symbols to tell Maude the criteria that must be met to stop the search. By default, in Maude, searches are performed on a state graph using the breadth-first search approach. The `search` command has the form "`search <Term-1> <SearchArrow> <Term-2>.`", where `<Term-1>` and `<Term-2>` are valid state statements of our modeled system. The word `<SearchArrow>` can be one of the following forms, each one giving a different criterion to proceed in the construction of the state graph:

- `=>1` to make a search of a single execution step, that is, only one rule is applied.
- `=>+` to make a search of one or more rewrite rules.
- `=>*` to make a search of zero, one or more rewrite rules.

```
rl [E1] :    p[E](pk(E,PK),pk(B,PK'),sk(E,SK),CSE) PS |
             msg(A -> E, {N,A}PK)
             =>
             p[E](N,pk(E,PK),pk(B,PK'),sk(E,SK),CSE) PS |
             msg(A -> B, {N,A}PK') .

rl [E2] :    p[E](CSE) PS | msg(B -> A, C)
             =>
             p[E](CSE) PS | msg(E -> A, C) .

rl [E3] :    p[A](pk(A,PK),sk(A,SK),N,CSA)
             PS | msg(E -> A, {N,C}PK)
             =>
             p[A](C,N,CSA) PS | none .

rl [E4] :    p[A](Nb,pk(E,PK),CSA) PS | none
             =>
             p[A](CSA) PS | msg(A -> E, {Nb}PK).

rl [E5] :    p[E](pk(E,PK),pk(B,PK'),sk(E,SK),CSE)
             PS | msg(A -> E, {N}PK)
             =>
             p[E](N,CSE) PS | msg(A -> B, {N}PK') .
```

**Figure 7** **Rules regarding the capabilities of the intruder as defined by the simplified version of the NSPK protocol.**

- =>! to make a search of only canonical states, in other words, a state to which no rules can be applied.

As a first example, we will check if we can reach, from a valid initial state, a state in our model where two participants have shared some information, *i.e.,* they have followed the protocol accordingly. Figure 8 is the output given by the command where init is a valid initial state previously defined. The result is that Maude finds two possible states where two honest participants have applied the protocol.

As a second example, we can try something more interesting. We will see if a participant manages to trick both sides in the system we have modelled, which is a critical situation an honest participant does not want to happen. Searches for undesired states that do not find solutions mean that the system's critical situation is not reached from the specified initial state. Figure 9 shows the use of the search command with the same initial state as before, but now it has a final state where the intruder has acquired both secrets from the other two participants.

The other path to formally specify and verify properties is using a technique called model-checking. Maude has a model checker (*Eker, Meseguer & Sridharanarayanan, 2004*) to perform model checking by writing properties as formulas in linear temporal logic

```
search in NSPK : init =>* (PS p[A](Na,Nb) p[B](Na,Nb)) | M:Messages .

Solution 1 (state 20)
states: 21  rewrites: 61 in 0ms cpu (1ms real) (102693 rewrites/second)
PS --> p[E](pk(A, Kpa),pk(B, Kpb),pk(E, Kpe),sk(E, Kse))
M:Messages --> none

Solution 2 (state 27)
states: 28  rewrites: 76 in 0ms cpu (3ms real) (79166 rewrites/second)
PS --> p[E](Na,pk(A, Kpa),pk(B, Kpb),pk(E, Kpe),sk(E, Kse))
M:Messages --> none

No more solutions.
states: 33  rewrites: 96 in 1ms cpu (6ms real) (70021 rewrites/second)
```

**Figure 8** Output of a search command to examine the execution paths for states where two participants have applied the NSPK protocol, and thus share their respective secret values.

```
search in NSPK : init =>* (PS p[E](Na,Nb,CSE)) | M:Messages .

Solution 1 (state 31)
states: 32  rewrites: 90 in 0ms cpu (6ms real) (119521 rewrites/second)
PS --> p[A](Na,pk(B, Kpb)) p[B](Na,Nb)
CSE --> pk(A, Kpa)
M:Messages --> none

No more solutions.
states: 33  rewrites: 96 in 1ms cpu (8ms real) (90566 rewrites/second)
```

**Figure 9** Output of a search command to examine the execution paths for a MITM attack in the NSPK protocol symbolic specification.

and using modules to describe predicates about the system states. We further explain this concept and put it into practice in 'Verification'.

## Functional modules

Functional modules specify functions and are similar to system modules, except that they can only contain equations. Functional modules, unlike system modules, are deterministic and finite. To define a functional module in Maude, we do so between the fmod and endfm keywords. Let us see an example of specifying a functional type module.

If we look at the following figure, we have a functional module, defined between fmod and endfm, which specifies through equations the factorial mathematical function of a number *N*. In the module, we have defined the symbol for said operation and the equations that will transform the expression until it is irreducible, thus obtaining the result.

Continuing with Fig. 10, on line three, we specify the symbol of the factorial function, which obtains a natural number in place of the _ symbol and returns another natural number, as indicated by the expression Nat − > Nat.

Then, on line four, we declare the variable N, which we will use in the following equations on lines five and six. The first equation defines that when Maude finds a "0 !" on the left

```
1  fmod FACT is
2      protecting NAT .
3      op _! : Nat -> Nat .
4      var N : Nat .
5      --- factorial for N=0 is 1
6      eq 0 ! = 1 .
7      --- factorial for N>0 is (N-1)! * N
8      eq N ! = (sd(N,1))! * N [owise] .
9  endfm
```

**Figure 10  Code of a functional module that implements the factorial in Maude.**

side, this is translated into the number one. The second equation will be executed whenever the first one has not executed thanks to the attribute defined *owise*, and it will return the factorial function of a number *N* as a factorial of *N-1* multiplied by *N*.

In addition, it should be noted that just as there are conditional rules, there are also conditional equations. These behave in the same way as the equations described, with the difference that one or more conditions must pass to be applied, in the same way as the conditions of the rules. The syntax for conditional equations is "ceq <Term1>= <Term2>if <Condition-1>/ ... / <Condition-k>.", where conditions <Condition-k> are expressions that will return false or true. The conditions of the conditional equations can take two forms:

1. Be other equations, with the form t=t'.
2. Matching equations, with the form t:=t'.

To finish, we will see the reduce command. This command is very useful for us to test the defined equations in functional or system modules. The reduce command receives an expression and returns the maximum reduced expression using the equations we have defined. For example, we will use the definition of the factorial operation in the FACT functional module to test different expressions. For example, as Fig. 11 shows, we tried to reduce the factorial of three and the factorial of zero. The first returns six, and the second returns one, both correct. In addition, since no rules are applied in reduce, we can see how in both executions, the number of rewrites is zero, indicating that no rule was applied and, therefore, the state was not changed.

# KEY ENCAPSULATION MECHANISMS

In this section, we explain three key encapsulation mechanisms: Kyber (*Avanzi et al., 2019*), BIKE (*Aragon et al., 2017*) and McEliece (*Chou et al., 2022*). Specifically, we will explain their behaviour and the security fundamentals behind each one of them to give the reader an understanding of what our work has been based on.

## Behaviour

The three key encapsulation mechanisms' behaviour fits the following model represented in Fig. 12. Participants of the protocol will be Alice and Bob for literature reasons. Before

```
[Maude> reduce 3 ! .
reduce in FACT : 3 ! .
rewrites: 10 in 0ms cpu (0ms real) (46082 rewrites/second)
result NzNat: 6
[Maude> reduce 0 ! .
reduce in FACT : 0 ! .
rewrites: 1 in 0ms cpu (0ms real) (1000000 rewrites/second)
result NzNat: 1
```

**Figure 11  Execution of two factorial expressions with the reduce command.**

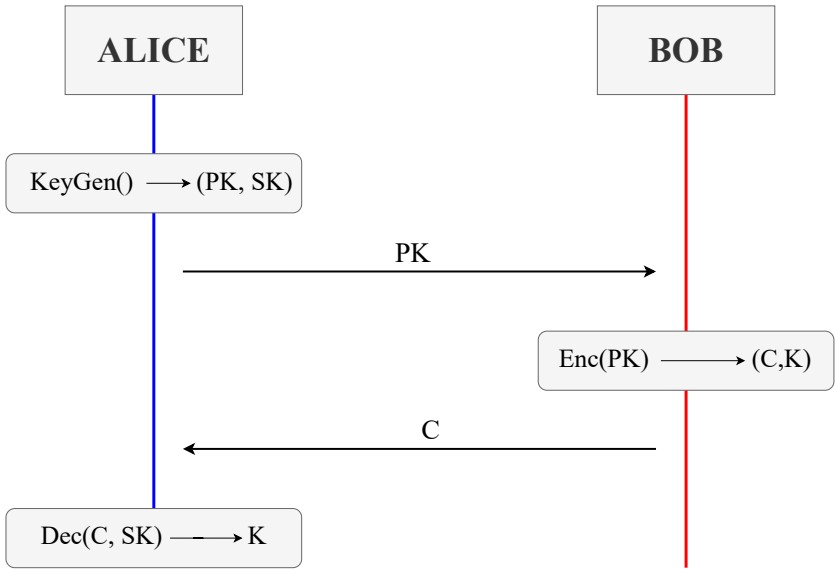

**Figure 12  High-level view of the behaviour of a key exchange of a KEM between two honest partici-pants, *i.e.,* Alice and Bob.**

explaining an example trace of the execution, we look into the three common steps in these KEMs.

- KeyGen() → (PK, SK): Step where a pair of values is generated, acting as public and secret keys during the session. Stands for Key Generation.
- Enc(PK) → (C, K): Step where using the public key PK of another participant, the values, that change depending on the KEM, used in the computation of a shared key K are encapsulated into a ciphertext C. Stands for encapsulation.
- Dec(C, SK) → K: Step where previously encapsulated value or values in the ciphertext C are now decapsulated using some operations alongside the secret key SK. These values are then used to compute the shared key K. Stands for decapsulation.

A KEM is initiated when a participant, *i.e.,* Alice, performs the KeyGen function, using some predefined values of the scheme, to generate a pair of keys (PK, SK). The former key, PK, is the public key and can be known by every other network participant. The latter,

SK, is a secret key only known and accessible by the user that generated it, in this case, Alice. Once Alice has both keys, she will send a message to another participant, Bob, with her public key. Once Bob receives Alice's public key, he performs Enc. The Enc function produces a pair (C, K), where C is a ciphertext encapsulating some key information for later deriving the second element of the pair, K, which is the future shared key between the two participants. Once Bob has in his storage the shared key and the ciphered text, he sends the latter in a new message back to Alice, who started the protocol session and from whom he used the public key. Alice then receives the ciphered text C from Bob and uses her secret key SK to perform the Dec function over ciphertext C. Function Dec outputs ideally the original encapsulated information, allowing for the computation of the same shared key generated by Bob. With the conclusion of this last step, participants have securely shared a key K between them.

As we have seen, the network is elementary, and participant interaction is minimal. No confirmation messages or previous establishment to know where the participants are in the network is performed. We then assume that any necessary discovery and set-up procedures have been done previously by the corresponding participants.

## Security fundamentals

In this section, we focus on the security aspects of each KEM. We recall the algebraic properties each KEM needs to function properly in order to be formally specified using our framework in 'Framework Specification' below.

### *Kyber*

Kyber (*Avanzi et al., 2019*) is an IND-CCA2-secure key encapsulation mechanism whose security is based on the hardness of solving the learning-with-errors (LWE) problem over module lattices. Kyber works with vectors and matrices of polynomials with various operations, such as concatenation, transposition, product or other more complex ones, such as hash and key derivation functions. These operations are present in the main functions KeyGen, Enc and Dec, depicted in Figs. 13 and 14. It is important to clarify that the algorithms in both figures represent an instance of the algorithms previously explained with Fig. 12 in 'Behaviour'. Specifically, Figure 13 shows three algorithms that encapsulate the ones present in Fig. 14. The encapsulation is done to provide further security measures as stated by *Avanzi et al. (2019)*. The algorithms on the left, KeyGen and Dec, are executed by the initiator of the protocol. Meanwhile, the algorithm on the right, Enc, is run by the other participant.

Furthermore, In Fig. 13 it should be noted that in algorithm **CCAKEM.Dec**, when the sub-function CPAPKE.Dec($\mathbf{s}$,($\mathbf{u}$,v)) takes place, the computed text $m'$ could not be the same as the one generated by the other participant in CCAKEM.Enc. This different message $m'$ is a value close to $m$ given the property *isSmall(p)* over a polynomial $p$. We say that a polynomial $p$ is *small* when its degree is lower than a given number established by the protocol. This approximately equal value $m'$ is then used to compute a new, but also close, value $c'$, which is compared to the received $c$ in a message. Depending on their equality, the construction of the shared key will be different from the key derivation function (KDF).

**CCAKEM.KeyGen** $: () \to pk, sk$
$1 : z \leftarrow \mathcal{B}^{32}$
$2 : (pk, sk') = \textbf{CPAPKE.KeyGen}()$
$3 : sk = (sk'||pk||\textbf{H}(pk)||z)$

$\xrightarrow{\quad pk \quad}$

**CCAKEM.Enc** $: pk \to c, K$
$1 : m \leftarrow \mathcal{B}^{32}$
$2 : m \leftarrow \textbf{H}(m)$
$3 : (\bar{K}, r) = \textbf{G}(m||\textbf{H}(pk))$
$4 : c = \textbf{CPAPKE.Enc}(pk, m, r)$
$5 : K = \textbf{KDF}(\bar{K}, \textbf{H}(c))$

$\xleftarrow{\quad c \quad}$

**CCAKEM.Dec** $: c, sk \to K$
$1 : pk = sk + 12 \cdot k \cdot n/8$
$2 : h = sk + 24 \cdot k \cdot n/8 + 38 \in \mathcal{B}^{32}$
$3 : z = sk + 24 \cdot k \cdot n/8 + 64$
$4 : m' = \textbf{CPAPKE.Dec}(\mathbf{s}, (\mathbf{u}, v))$
$5 : (\bar{K}', r') = \textbf{G}(m'||h)$
$6 : c' = \textbf{CPAPKE.Enc}(pk, m', r')$
$7 : \textbf{if } c = c' \textbf{ then } K = \textbf{KDF}(\bar{K}||\textbf{H}(c))$
$\qquad \textbf{else } K = \textbf{KDF}(z||\textbf{H}(c))$

**Figure 13  Algorithms of Kyber Key Encapsulation Mechanism adapted from *Avanzi et al. (2019)*.**

**CPAPKE.KeyGen** $: () \to pk, sk$
$1 : \mathbf{A} \leftarrow R_q^{k \times k}$
$2 : \mathbf{s} \leftarrow R_q^k$
$3 : \mathbf{e} \leftarrow R_q^k$
$4 : pk = \mathbf{As} + \mathbf{e}$
$5 : sk = \mathbf{s}$

$\xrightarrow{\quad pk \quad}$

**CPAPKE.Enc** $: pk, m, r \to pk, sk$
$1 : \mathbf{A} \leftarrow R_q^{k \times k}$
$2 : \mathbf{r} \leftarrow R_q^k$
$3 : \mathbf{e}_1 \leftarrow R_q^k$
$4 : e_2 \leftarrow R_q^k$
$5 : \mathbf{u} = \mathbf{A}^T \mathbf{r} + \mathbf{e}_1$
$6 : v = \mathbf{t}^T \mathbf{r} + e_2 + \texttt{Decompress}_q(m, 1)$
$7 : c = (\texttt{Compress}_q(\mathbf{u}, d_u), \texttt{Compress}_q(v, d_v))$

$\xleftarrow{\quad c \quad}$

**CPAPKE.Dec** $: sk, c \to m$
$1 : \mathbf{u} = \texttt{Decompress}_q(c, d_u)$
$2 : v = \texttt{Decompress}_q(c, d_v)$
$3 : \mathbf{s} = sk$
$4 : m = \texttt{Compress}(v - \mathbf{s}^T \mathbf{u}, 1)$

**Figure 14  Internal algorithms of the Kyber Key Encapsulation Mechanism adapted from *Avanzi et al. (2019)*.**

This differentiation of values arises with low probability (*Avanzi et al., 2019*), but it states that the encryption and decryption phases are not error-prone.

Nevertheless, why are they different in the end? To answer this question is to understand the strength of the scheme against a quantum adversary. If we check the most internal functions, that is, the CPAPKE ones shown in Fig. 14, we can see in algorithm CPAPKE.Enc that there are vector values such as $\mathbf{e}_1$ and $e_2$ that have been sampled with a random seed

$$\mathbf{KeyGen} : () \rightarrow (h_0, h_1, \sigma), h$$
$$1 : (h_0, h_1) \xleftarrow{\$} H_w$$
$$2 : h \leftarrow h_1 h_0^{-1}$$
$$3 : \sigma \xleftarrow{\$} \mathcal{M}$$

$\xrightarrow{\quad h \quad}$

$$\mathbf{Encaps} : h \rightarrow \mathbf{K}, c$$
$$1 : m \xleftarrow{\$} \mathcal{M}$$
$$2 : (e_0, e_1) \leftarrow \mathbf{H}(m)$$
$$3 : c \xleftarrow{\$} (e_0 + e_1 h, m \oplus \mathbf{L}(e_0, e_1))$$
$$4 : K \leftarrow \mathbf{K}(m, c)$$

$\xleftarrow{\quad c \quad}$

$$\mathbf{Decaps} : (h_0, h_1, \sigma), c \rightarrow K$$
$$1 : e' \leftarrow decoder(c_0 h_0, h_0, h_1)$$
$$2 : m' \leftarrow c_1 \oplus \mathbf{L}(e')$$
$$3 : \textbf{if } e' = \mathbf{H}(m') \textbf{ then } K \leftarrow \mathbf{K}(m', c)$$
$$\textbf{else } K \leftarrow \mathbf{K}(\sigma, c)$$

**Figure 15  Algorithms implementing the three main steps of the KEM BIKE, adapted from *Aragon et al. (2017)*.**

$r$ using function sampleCBD from a centred binomial distribution. These values add randomness to the computations, thus making the computations harder to replicate or break.

$$Decompress_q(Compress_q(X, 1), 1) = X' \tag{1}$$

Function $Decompress_q$ processes the error when extracting from the pair of vectors $c_1$ and $c_2$ the vectors $\mathbf{u}$ and $v$ respectively, at the middle of step CPAPKE.Dec. This function has a property in combination with function $Compress_q$, which states in Eq. (1) that decompressing the compress of a given value $X$ with the same second parameters, gives a new value $X'$ which is similar to the original compressed value. This property takes place while computing the original message m as it is shown in Eq. (2), present at the end of the Dec algorithm in Fig. 14.

$$m = Compress(v' - s^T \mathbf{u}', 1) \tag{2}$$

It is important to mention that other operations take place, such as *generate*, *sampleCBD*, *Compress*, *Decompress*, *encode* and *decode*. These are necessary for the main three operations we have described before but are not explained in detail in this article because they are not necessary for the understanding of the protocol. Full descriptions of these specific functions are available at *Avanzi et al. (2019)*.

### *BIKE*

BIKE (*Aragon et al., 2017*), which stands for BIt-flipping Key Encapsulation, is a code-based key encapsulation mechanism. Figure 15 shows BIKE's specific algorithms. Although the previous KEM Kyber is lattice-based and this KEM BIKE is code-based, they follow a similar structure, and BIKE's algorithms are an instance of the functionalities of 'Behaviour'. The key part of this KEM is how the decapsulation algorithm, to process the received ciphertext, manages to recover the original tuple of errors computed during the

encapsulation step. To better understand the process, we first need to know where the errors are added. Looking at Fig. 15 we can see in step *Encaps*, line 3, that $C$ is computed as a tuple of values $(c_0, c_1)$. The first element, $c_0$, is defined as $e_0 + e_1 h$, being $h = h_1 h_0^{-1}$ the public key previously received and $(e_0, e_1)$ the tuple of error values we are looking for. The second element, $c_1$, encapsulates the value $m$ performing the exclusive or operation with a hash function $L : \mathcal{R}^2 \to \mathcal{M}$ on the two same error values. This value $m$ holds significant importance since it is used to compute, through other hash function $H : \mathcal{M} \to \epsilon_t$, the pair of error values, $(e_0, e_1)$, specifically in line 2.

Now, on the *Decaps* step line 1, present also in Fig. 15, $e'$ will be the result of applying the *decoder* function on $c_0$ with the components of the public key $h$, that is, the tuple $h_0$ and $h_1$. This decoding algorithm will ideally return a pair of errors equal to those of $e$ from the *Encaps* step such that $e_0 h_0 + e_1 h_1 = s$, where $s$ represents the first parameter of *decoder*. Since we managed to extract the original errors with high probability, we are able to compute $m'$ on line 2 applying exclusive or over $c_1 = m \oplus L(e_0, e_1)$ and $L(e_0, e_1)$. It is important to note that similar to the previous KEM, $m'$ here is named to denote that the computed value may not be the same since the *decoder* function might fail with low probability. Thanks to two properties of exclusive or, we will get the original $m$. The first property is where $X \oplus X = 0$, and the second property has to do with $X \oplus 0 = X$. The combination of these two properties allows us to compute in line 3 the same shared key using another hash function $K$.

Another operation also takes place and is fully defined by *Aragon et al. (2017)*. The concrete algorithm is *Black-Gray-Flip*, and it implements the decoder. It is necessary for the decoding of a ciphertext, but we do not explain it in detail because it is not necessary for the understanding of the protocol.

### Classic McEliece

Classic McEliece (*Chou et al., 2022*) is a code-based key encapsulation mechanism designed to achieve IND-CCA2 security. Even though it is code-based, like BIKE, and follows the same steps all KEMs do, it is a new instance, providing new algorithms for each of the three main functions for key generation, encapsulation and decapsulation.

The key part for this specific instance of a KEM resides at the *ENCODE* and *DECODE* functions, and the role they play in the encapsulation and decapsulation algorithms, respectively. Figure 16 shows, in algorithm *Enc* line 2, the application of function *ENCODE* over some error $e$ with the public key $T$ to get the ciphertext $C$. A careful reader can see the resemblance between BIKE's encapsulation and this one because some sampled or generated error is being "encapsulated" with the public key previously received. Then the shared key is computed using a hash function $H : \{0, 1\}^8 \times \mathbb{F}_2^n \times \mathbb{F}_2^{mt} \to \mathbb{F}_2^l$.

In algorithm *Dec*, values $s$ and $\Gamma'$ are extracted from the private key computed with *SeededKeyGen*, that is, *KeyGen*. With these values, the application of function *DECODE* manages to decapsulate the original error $e$. This is done thanks to the idea represented in Eq. (3). Furthermore, notice that even if the decoding process can fail, *i.e.*, returning $e$ as $\perp$, there is a way in which the shared key can be computed thanks to the existence of some

**SeededKeyGen** $: \delta \rightarrow T, (\delta, c, g, \alpha, s)$
$1 : E = G(\delta)$
$2 : \delta' \leftarrow_l E$
$3 : s \leftarrow_n E$
$4 : \alpha_0, \ldots, \alpha_{q-1} = \texttt{FIELDORDERING}(E)$
$5 : g = \texttt{IRREDUCIBLE}(E)$
$6 : \Gamma = (g, \alpha_0, \alpha_1, \ldots, \alpha_{n-1})$
$7 : (T, c_{mt-\mu}, \ldots, c_{mt-1}, \Gamma') \leftarrow \texttt{MATGEN}(\Gamma)$
$8 : \Gamma' = (g, \alpha'_0, \alpha'_1, \ldots, \alpha'_{n-1})$
$9 : c = (c_{mt-\mu}, \ldots, c_{mt-1})$
$10 : \alpha = (\alpha'_0, \alpha'_1, \ldots, \alpha'_{n-1}, \alpha_n, \ldots, \alpha_{q-1})$

$\xrightarrow{\hspace{1.5cm} T \hspace{1.5cm}}$

**Encap** $: T \rightarrow C, K$
$1 : e = \texttt{FIXEDWEIGHT}()$
$2 : C = \texttt{ENCODE}(e, T)$
$3 : K = \texttt{H}(1, e, C)$

$\xleftarrow{\hspace{1.5cm} C \hspace{1.5cm}}$

**Decap** $: C, (\delta, c, g, \alpha, s) \rightarrow K$
$1 : \Gamma' = (g, \alpha'_0, \alpha'_1, \ldots, \alpha'_{n-1}) \leftarrow s$
$2 : e = \texttt{DECODE}(C, \Gamma')$
$3 : K = \texttt{H}(1, e, C)$

**Figure 16** **Algorithms of McEliece adapted from** *Chou et al. (2022).*

error correction mechanisms.

$$DECODE(ENCODE(E, T), \Gamma') = E \tag{3}$$

It is important to note that although the second argument of *DECODE* and *ENCODE* are different, they share a common value. If we expand the values with their specific construction definitions, *i.e.,* the specific functions used in their computations, we get that $T$ is the first component from the output of *MATGEN* $(\Gamma)$ while $\Gamma'$ is the last component from the same output. With this discovery, we get a relation between the two, in principle, distinct values, so when the same matrix generation function has been used to compute the two values, the property defined in Eq. (3) holds.

As with previous KEMs, other operations take place. For Classic McEliece it is MatGen, Encode, Decode, Irreducible, FieldOrdering and FixedWeight. These are necessary for the main three operations we have described before but are not explained in detail here because only a high-level representation is needed for the specification. Full descriptions of these specific functions are available at *Chou et al. (2022)*.

## FRAMEWORK SPECIFICATION

This section presents the framework for the symbolic specification of KEMs. First, general and KEM-specific assumptions are explained. These assumptions are necessary to cover the mathematical notions the symbolic model cannot manage, thus abstracting ourselves from computational problems. Then, an overview of the framework is given.

## Assumptions

During the design of our framework, assumptions on the symbolic models are made to ease the specification process by allowing us to abstract from implementation details. Under such assumptions, systems are easier to specify and understand, assisting in the automatization process by allowing the engineer to focus his efforts on capturing the behaviour and key transformations of the KEM. We have taken the freedom to make some assumptions on the specification of the symbolic modules for the three KEMs in our framework.

The assumptions are divided into Dolev-Yao adversary assumptions and KEM-specification assumptions. The former assumptions, Dolev-Yao, are stated next, implying additional rules and conditions for our Maude system module. These new rules are explained in 'Framework Infrastructure'. We also assume that there will be only three participants in the network, two of them honest (Alice and Bob) and one adversary (Eve). The latter assumptions are almost all over mathematical and low-level concepts for each KEM. Such assumptions allow us to abstract our model from implementation requirements and focus on representing the desired behaviour among the participants, only specifying the essential part of these mechanisms based on the explanations in 'Security Fundamentals' for each of them.

### Dolev-Yao

The Dolev-Yao adversary model was first introduced by *Dolev & Yao (1983)*. In that article, the authors explained that public-key schemes are secure against adversaries that cannot modify the environment, which is unrealistic. That is why they presented different examples of protocols whose security properties could be compromised if an intruder can take action over the messages of a network. An intruder can be either passive or active over a network where other participants send and receive messages during, for example, a handshake protocol or a key exchange scheme. The passive intruder can only read the message and extract raw content from it, meaning they cannot derive any information from messages without the proper private key. The active intruder can read messages, modify them, and send them through the network. It is essential to clarify that the intruder is considered a polynomial-time Turing machine.

In their seminal work, the authors proposed the Dolev-Yao intruder model. This model states the capabilities an intruder has over a network. Such capabilities are:

- Intruder can obtain any message that is passing through the network.
- He is a *legitimate* user of the network. That is, he can do any actions an honest participant can.
- The intruder has the opportunity to be a receiver to any participant. That is, he can receive messages from other participants.

It must be noted that the network participants, the intruder included, must comply with the following assumptions:

- One-way functions are unbreakable. In other words, the basic primitives of the protocol are considered to be non-reversible.

- The protocol definition cannot be changed and must be followed by any participant. A user cannot do undefined steps during the protocol execution.
- Public keys can be used for encryption by everyone.
- Private keys can decrypt messages encrypted by the corresponding public key.

### *Kyber*

In the case of the KEM known as Kyber, we make some assumptions on the mathematical aspects of the data types we try to represent. We also assume certain qualities of some operations. In the case of basic data types, we assume that all matrices are square matrices and that vectors are column vectors, considering transposed vectors as row vectors. Vectors, representing polynomials, are considered to be of the necessary degree to be considered small, thus fulfilling the property *isSmall(p)* explained in 'Kyber'. Related to the qualities of some operations, we only consider for the decompression function the ideal case where there is no error in obtaining $m$. Thus, $m'$ will be equal to $m$. Furthermore, we assume that the deciphered message is the shared key between the participants, so no additional functions, such as KDF, need to be specified and applied. Finally, our sampling procedures are deterministic, but we will assume that the operators sampled are from a CBD whenever it is the case.

### *BIKE*

Regarding the assumptions made over the symbolic model of BIKE, we have some inherited from the specification provided by *Aragon et al. (2017)* and the rest are made over operation *decoder*. From the specification, we inherit the assumption for Hardness of Quasi Cyclic Syndrome Decoding (QCSD) and Hardness of Quasi Cyclic Codeword Finding (QCCF), needed for code-based KEM. We also assume indistinguishability in two aspects. First, we assume it for the public key, *i.e.,* $h_1 h_0^{-1}$ from random $(h_o, h_1) \xleftarrow{\$} \mathcal{H}_w$. Second, assume indistinguishability of the first half of the ciphertext $c$, *i.e.,* $(e_0 + e_1 h)$ where $h = h_1 h_0^{-1}$, from random $((e_0, e_1), h) \xleftarrow{\$} \varepsilon_t \times \mathcal{R}$. We assume the *decoder* function is correct and always returns the correct pair of errors, so we assume no decoding failure, *i.e.,* $\sigma = 0$.

### *Classic McEliece*

With Classic McEliece, we focus on operational assumptions regarding its decoding algorithm DECODE. As per the specification given by *Chou et al. (2022)*, algorithm DECODE has two possible outputs. On the first one, the original errors are recovered given some conditions, *i.e.,* there exists a weight-t vector $e \in \mathbb{F}_2^n$ such that $C = He$ with $H = (I_{mt} | T)$, then DECODE $(C, \Gamma') = e$. The second condition is when the ciphertext does not have the form $He$, so DECODE returns DECODE $(C, \Gamma') = \bot$. For this reason, we assume that all applications of DECODE always return the original errors; thus, $C$ always has the form $He$ for any weight-t vector $e \in \mathbb{F}_2^n$.

## Framework infrastructure

Different modules carry on the specification of our framework. Each of these modules is specific to one part of the KEMs, and, when set together, builds a representative symbolic model. In this subsection, we will examine them and explain the basic contents and their

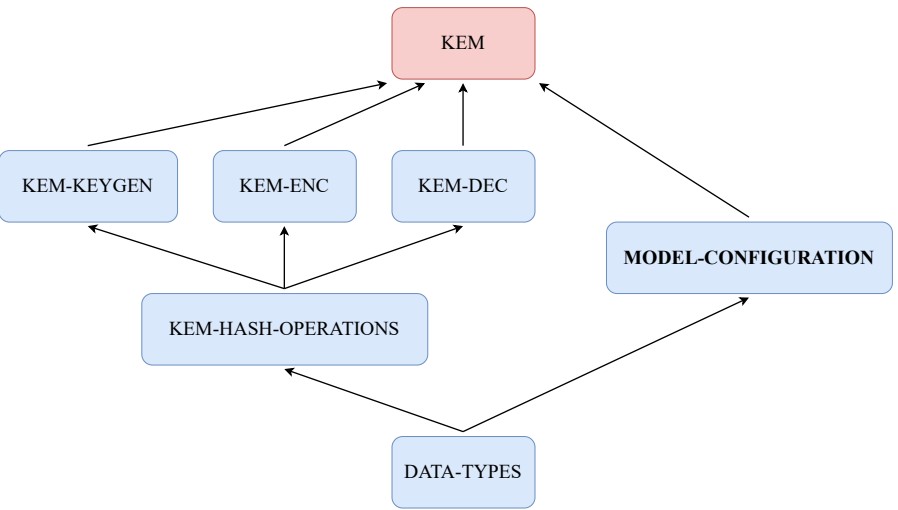

**Figure 17** **Structure of our framework in modules with relations of inclusion between them.**

purpose. Specifically, all KEMs in our framework are specified with six functional modules and one system module. An overview of the structure of the framework is presented in Fig. 17. Boxes named DATA-TYPES, KEM-HASH-OPERATIONS, KEM-KEYGEN, KEM-ENC, KEM-DEC and MODEL-CONFIGURATION represent functional modules, where KEM is the corresponding name of the KEM, *i.e.,* for KYBER we have KYBER. The system module is named after the corresponding KEM following the previous example.

### Functional modules

*DATA-TYPES.* Basic data types and operations over them are specified in this module. Data types are defined with *sorts*, and relations between them are defined with the *subsort* relation. For example, in most of the KEMs, the basic data type is polynomials, and one fundamental form of a polynomial might be a natural number, between others. Operations over the basic data types are defined here *via* operator declaration symbols. These operators might have some properties defined through the use of axioms like associativity, commutativity and/or the identity element. Other properties, like the distributive one, can be specified through the appropriate equations to define properties between operations. Another important feature is that most assumptions of a protocol, like the ones mentioned in 'Assumptions' are portrayed here. For example, if in a KEM there is an inverse notion, then it is interesting to specify the negation case where something and its inverse give the identity element through some operation. These assumptions and properties will help in the resolution of the model, making things simpler as we will see in subsequent modules.

Furthermore, complex data structures like pairs and lists of data are defined here. For example, the data structure `Pair` needs the declaration of some function which can access its data elements. To this end, along the definition of a `Pair`, functions `first` and `second` are defined over pairs to access the first and second components, respectively.

*KEM-HASH-OPERATIONS.* The module has the responsibility of declaring and defining any hashing functions present in the KEM. These hash functions are declared using the operator declaration syntax of Maude, and the semantics is defined using equations. Such a definition is symbolic and deterministic, representing the values that would come from the application of the hash to a symbolic value.

*KEM-KEYGEN.* The key generation module aims to harbour the sampling operations, values and functions necessary to generate pairs of public and private keys. These sampling functions would assume the corresponding sampling distributions and be modelled through equations to represent the sampling procedure symbolically.

*KEM-ENC.* Following a similar philosophy from the previous key generation module here would be defined the necessary samplings, with their assumed distributions, and cryptography operations. As always, all of them are defined in a symbolic manner. In the end, all this would represent the necessary components to carry on with the encapsulation process of some value as a ciphertext.

*KEM-DEC.* The key to the specified KEMs resides in this module. Here the equational theories related to the core operations are declared and defined. These equations represent properties in the underlying operations, be they mathematical or not. A simple example to illustrate a property over some operation would be the well-known Diffie Hellman key exchange algorithm. At the end of the algorithm, both participants derive the same values thanks to the property of modular exponentiation such that $(g^a \bmod p)^b \bmod p = (g^b \bmod p)^a \bmod p$.

*MODEL-CONFIGURATION.* Module that sets the basic structure of the system upon which the KEM will work on. We first define the participants, explaining their structure and components. Also, related to their components, we define some notation so participants can store and identify different types of keys and encrypted texts, linking those to a participant. Then the message syntax is set. Finally, we define the global state, where participants send and receive messages. All these element definitions are the same for the three KEMs we have defined following the framework.

Participants in our model follow the operational definition shown in Fig. 18. Here three identifiers are declared for our corresponding participants. Then the structure of a participant is declared. A participant consists of an *Identifier*, like the ones that have been defined, a group of keys the participant know, and a group of elements that represent its elements that are not keys and the participant has in its possession, that is, in its memory.

Shared between all KEMs, these operators have the purpose of storing a relation between a cryptography component and a participant. Examples of such cryptography components are the notion of types of keys such as public, secret and shared keys, or a ciphertext. Figure 19 shows the definition of the former, where sorts PKey and SKey represent the publicly available keys and secret keys, only known to the creator and or corresponding peer, respectively.

```
ops Alice Eve Bob : ->  Identifier .
op _[_]_ : Identifier Keys Content -> Principal .
```

**Figure 18**  Definition of a participant at the network in `MODEL-CONFIGURATION`.

```
op publicKey : Identifier Key -> PKey .
op secretKey : Identifier Key -> SKey .
op sharedKey : Identifier Key -> SKey [format (g! o)] .
```

**Figure 19**  Definition of components to link three notions of keys with a participant.

```
op msg{(_,_)[_]_} : Identifier Identifier MsgState Content -> Msg .
```

**Figure 20**  Definition of the message operator for our symbolic model.

Messages are defined by the operator *msg* shown in Fig. 20. A message contains information about two participant identifiers, the status of the message and the content it carries. The first identifier indicates the source of the message, and the second is the identifier of the participant to whom the message is delivered throw the network. Then, the status of a message can take the values *sentX* and *receivedX*, where *X* can either be *PK* or *C* for public key or ciphertext, respectively, depicting in part the current step over the network. At last, the content of the message is assumed to be secure, meaning a participant can not infer any additional contents from it without the required information.

Figure 21 shows the definition of the structure representing our system. Here, and from left to right, the elements our rules will handle are specified. At the right corner, we assigned a field between symbols { and } for all the available sample values. These samples are modelled as constant values representing symbolic values, to use in any of the *Keygen*, *Enc* or *Dec* steps. Next, we have an associative and commutative set of participants between symbols < and >, following the structure explained at the beginning of this section. Then, at the left end, we have the network, with a collection of associative messages representing a kind of record that lets participants work over the sent messages. All this together gives us a proper term structure that can represent any network with any participants and messages being sent and received.

### System module

With the system module identified as the name of the corresponding KEM, we try to model the behaviour of honest participants and the intruder's capabilities over the network, that is, the global state we defined. The first thing to do in this module is to declare the sampling sets of values needed to perform a session of the key exchange. Once the samples are defined, we can start defining the honest participant behaviour, following the explanation in 'Behaviour'. These rules specify the transitions of our formal system regarding the three main functions known as *KeyGen*, *Enc* and *Dec*, and also model the sending and receiving

```
op {_}<_>net(_) : Content Principals Msgs -> GlobalState .
```

**Figure 21  Definition of the syntax to represent the global state of our system.**

```
ops init1 : -> GlobalState .
eq init1 = {ds(d1) ms(m1) rs(r1)}
            < (Alice[emptyK]peer(none))
              (Eve[emptyK]peer(none))
              (Bob[emptyK]peer(none)) >
            net(emptyM)  .
```

**Figure 22  Example of initial state definition for KYBER.**

of both the public key and the ciphered text over the network. Lastly, we check the specific intruder behaviour following the Dolev-Yao intruder model assumptions and represent such capabilities over the network messages.

**Initial set up:** Before modelling the behaviour, we need to declare the sampling groups and the initial states upon which the symbolic model would start to run the simulation. Sampling groups are represented as operators that can store sampling values of some type. Such operators are placed at the sample set of the global state. The population in these operators should be minimal, that is, the minimal set of values required to carry on a protocol session of a KEM. In this way, and since our approach is symbolic, with many different symbolic values, we would be representing the same thing, so by keeping it minimal, we are avoiding redundancy. We can now declare the initial global states with the specific sampling sets, participants and the network channel. Such initial states can be seen as snapshots of the network. An example is shown in Fig. 22 following the required samples for KYBER, where sampling sets *ds*, *ms* and *rs* contain one corresponding sampling value, participants *Alice*, *Bob* and *Eve* have empty key sets and no peer assigned, and the network is empty.

**Participant behaviour:** Following the different KEMs' specifications and the Dolev-Yao assumptions, we specify the following rules in Maude to model how the different KEMs operate. All these rules have been written to be as general as possible, making the model and the constructed execution tree more realistic and compelling for model checking. It is important to note that the main differences between each rule related to the three main functions are the conditions, which help to model how the sample values and other functions are used to construct the cryptography elements like keys and ciphered texts between others. To properly explain the participant behaviour rules, we will first show a more general rule without the conditionals, which is almost the same for all the KEMs. Then, we will show the conditionals of each KEM and link it to the original algorithm we showed in 'Security Fundamentals'.

```
crl [KeyGen] :  {sampleSet(SAM1 CONT1) CONT4}
                < (ID1[emptyK]peer(none)) PS >
                net(MSGS)
                =>
                {sampleSet(CONT1) CONT4}
                < (ID1[publicKey(ID1, PK) ; secretKey(ID1, SK)]
                   sampleI(ID1, SAM1) peer(none)) PS >
                net(MSGS)
```

**Figure 23   General definition of conditional rule KeyGen in our framework.**

```
crl [SendPK] :  {CONT4}
                < (ID1[publicKey(ID1,PK) ; KS1]dI(ID1,SAM1) peer(none) CONT1)
                  (ID2[KS2]peer(none)) PS >
                net(MSGS)
                =>
                {CONT4}
                < (ID1[KS1]dI(ID1,SAM1) peer(ID2) CONT1)
                  (ID2[KS2]peer(none)) PS >
                net(MSGS msg{(ID1,ID2)[sentPK]PK})
                if (msg{(ID1,ID2)[sentPK]PK}) in MSGS == false /\
                   (msg{(ID1,ID2)[receivedPK]PK}) in MSGS == false /\
                   (ID1 =/= ID2) .

rl [ReceivePK] :    {CONT4}
                    < (ID2[KS2]peer(none) CONT2) PS >
                    net(MSGS msg{(ID1,ID2)[sentPK]PK})
                    =>
                    {CONT4}
                    < (ID2[publicKey(ID1,PK) ; KS2]peer(ID1) CONT2) PS >
                    net(MSGS msg{(ID1,ID2)[receivedPK]PK}) .
```

**Figure 24   Definition of rules SendPK and ReceivePK to send and receive a public key in our framework.**

The first rule is *KeyGen* as seen in Fig. 23. This rule is the one that starts the protocol for a given participant. Specifically, the rule states that given a participant with identifier ID1 whose set of keys is empty and has no peer associated, he can generate a publicKey(ID1, PK) and a secretKey(ID1, SK) in the group of keys, linked to the identifier *ID*1 of the participant. Specifically, for each protocol, the necessary samples should be present in CONT4, the pool of available sampled sets.

We also defined two rules available in Fig. 24 .The first rule, SendPK, models the behaviour of a participant with his public key, sending it to any other participant in the network different from him. The message is sent if it has not been sent previously, so we avoid infinite execution. Then, to complement the first rule, we defined rule ReceivePK to process an incoming message if it contains a public key and has not been received yet.

Rule *Enc*, as it is shown in Fig. 25, models the function with its same name. In order to apply the encryption step, a participant first has to receive the public key from the other peer, which is why the left-hand side of the rule has a component publicKey(ID1, PK). The participant also needs to be able to sample the necessary values from the set of samples

```
crl [Enc] : {CONT4}
              < (ID2[publicKey(ID1, PK) ; KS2]
                 peer(ID1) CONT2) PS >
           net(MSGS)
           =>
           {CONT4}
           < (ID2[sharedKey(ID1, SK) ; KS2]
              cI(ID1,C) peer(ID1) CONT2) PS >
           net(MSGS)
```

**Figure 25** Definition of conditional rule Enc in our framwork.

```
crl [SendCiph] :   {CONT4}
                   < (ID2[sharedKey(ID1,SK) ; KS2]
                      peer(ID1) cI(ID1,C) rI(ID1,SAM2) CONT2) PS >
                   net(MSGS)
                   =>
                   {CONT4}
                   < (ID2[sharedKey(ID1,SK) ; KS2]
                      peer(none) rI(ID1,SAM2) CONT2) PS >
                   net(MSGS msg{(ID2,ID1)[sentC](C)})
                   if  (msg{(ID2,ID1)[sentC](C)}) in MSGS == false /\
                       (msg{(ID2,ID1)[receivedC](C)}) in MSGS == false .

rl [ReceiveCiph] : {CONT4}
                   < (ID1[KS1]CONT1) PS >
                   net(MSGS msg{(ID2,ID1)[sentC](C)})
                   =>
                   {CONT4}
                   < (ID1[KS1]cI(ID1,C) CONT1) PS >
                   net(MSGS msg{(ID2,ID1)[receivedC](C)}) .
```

**Figure 26** Definition of rules SendCiph and ReceiveCiph to send and receive a ciphertext in our framework.

CONT4, just like in rule KeyGen. After applying the rule, the participant possesses in the pool of keys a new shared key SK related to his current peer ID1. In the content pool, the ciphertext C is stored as the value to be transmitted securely to the other participant.

The counterpart of SendPK but for a ciphered text *c* obtained through Enc is the conditional rule SendCiph. As Fig. 26 shows, it checks similar conditions when sending the public key, so no infinite execution happens. Then, the rule to receive a ciphertext behaves similarly to its counterpart, that is, ReceivePk. The rule is applied when there is a sent message in the network for a given participant with ID1. The content of the message is stored by the participant in its pool of content, acting as memory.

Finally, the rule to decipher the received ciphertext is Dec. Figure 27 depicts the behaviour of a participant ID1 with his own secret key SK and an assigned peer ID2. Once the shared key K1 for participant ID2, that is the peer, has been computed and placed in the pool of keys, the peer can be removed since the protocol session has ended between them.

**Intruder behaviour:** When specifying the intruder's capabilities over our module, we decided to specify two rules, Intercept1 and Intercept2, both identical between all the

```
crl [Dec] : {CONT4}
              < (ID1[secretKey(ID1,SK) ; KS1]
                peer(ID2) cI(ID1,C) CONT1) PS >
            net(MSGS)
            =>
            {CONT4}
            < (ID1[sharedKey(ID2,K1) ; KS1]peer(none) CONT1) PS >
            net(MSGS)
```

**Figure 27** **Definition of conditional rule Dec in our system module KYBER.**

```
crl [Intercept1] :  {CONT3}
                    < (Eve[publicKey(Eve, PK') ; KS1]peer(none) CONT1) PS >
                    net(MSGS msg{(ID1,ID2)[sentPK](PK)})
                    =>
                    {CONT3}
                    < (Eve[publicKey(ID1,PK) ; KS1]peer(ID1) CONT1) PS >
                    net(MSGS msg{(ID1,ID2)[sentPK](PK')})
                    if  ID1 =/= Eve /\
                        ID2 =/= Eve /\
                        ID1 =/= ID2 /\
                        PK =/= PK' /\
                        (publicKey(ID1,PK)) in KS1 == false .
```

**Figure 28** **Definition of conditional rule *Intercept1* in our system module KYBER.**

KEMs' symbolic models. The former can be seen in Fig. 28, and it binds the intruder with the ability to intercept a sent message containing a public key. The intercepted message is modified by extracting the body, that is the public key It is carrying, and replacing it with its own public key. This modification makes the receiver think the public key received is from the sender when it is not, thus beginning the man-in-the-middle attack. The latter is available in Fig. 29 and makes the intruder intercept a message sent with a ciphertext. This intercepted message is sent by the receiver from the previous fake message and makes Eve send a new message but with his own ciphertext. In this way, Eve has in store two ciphertexts, her own and the one intercepted.

# KEY ENCAPSULATION MECHANISM SPECIFICATIONS

Now we will look into the underlying equational theories and conditions for the rewrite rules of each KEM. On the equational theories, we refer to them as the equations defined in the functional modules that make the matching equations from the rules work. On the rewriting conditions, we refer to the aforementioned matching equations present in the conditions of the rewrite rules. The specific conditions can relate very closely to the specific steps or lines on each algorithm presented in 'Security Fundamentals'. First, we focus on Kyber, then pass on to BIKE, and finally, we explain the case of Classic McEliece.

```
crl [Intercept2] :   {CONT3}
                     < (Eve[KS1]cI(ID2,Cs') peer(ID2) CONT1) PS >
                     net(MSGS msg{(ID1,ID2)[sentC](Cs)})
                     =>
                     {CONT3}
                     < (Eve[KS1]cI(Eve,Cs) peer(ID1) CONT1) PS >
                     net(MSGS msg{(ID1,ID2)[sentC](Cs')})
                     if  ID1 =/= Eve /\
                         ID2 =/= Eve /\
                         ID1 =/= ID2 /\
                         Cs =/= Cs' /\
                         (cI(ID1,Cs)) in CONT1 == false .
```

**Figure 29  Definition of conditional rule *Intercept2* in our system module KYBER.**

## Kyber specification

Related to the properties explicitly shown in 'Kyber', we can model these properties using equations. In this case, the target module relates to the decapsulation, KYBER-DEC. This module is where the decompression of a compressed value must take place, and the error cancellation plays a role in removing the errors that add noise and makes the KEM difficult to break for an adversary.

The property between operations Compress and Decompress was presented as a key pillar to this KEM in 'Kyber' with Eq. (1). To model the relation, we defined equations
```
eq Decompress(Compress(X,N),N) = X .
eq Compress(Decompress(X,N),N) = X .
```
where X is a vector and N is a natural number. The two equations define the commutativity of their relation, which is necessary because the protocol applies Compress over Decompress.

Now, related to the property of error cancellation, since KYBER is a lattice-based KEM, we also represent in this module the process of eliminating the error from the ciphertext in order to obtain the message and finally derive the key. The concrete equation is
```
eq (V1 v+ Decompress(X,N)) v- V2 = Decompress(X,N) .
```
where X and N are the same from the previous property, and V1 and V2 are vectors. With this equation, we represent the ideal case where errors from $u'$ and $v'$ are properly cancelled because of subtraction leaving alone the Decompress of message m, necessary to compute the shared key. A more specific picture can be seen in the following way: $Compress(((tr + e_2) + Decompress(m, 1)) - (s^T(A^T r + e_1)), 1) \downarrow_E Compress(Decompress(m, 1), 1)$. The left-hand-side term is the complete instanced line four from step Dec in Fig. 14. Inside this term, we can see that variables V1 and V2 will be instantiated into $(tr + e_2)$ and into $(s^T(A^T r + e_1))$ respectively. We get the right-hand side term when reducing the term with equations, *i.e.*, the equation for error cancellation.

*KeyGen.* Regarding the construction of public and secret keys, this is done through the matching equations in the rule's conditions available in Fig. 30. The structure is the one present at the specification and previously depicted in Fig. 14, where public key PK is the

```
if  SK := sampleS(second(G(SAM1))) /\
    PK := ((generateA(first(G(SAM1))) m* SK)
            v+
            sampleE(second(G(SAM1)))) .
```

**Figure 30** Conditions of rule KeyGen in the symbolic model of Kyber.

```
if  ID1 =/= ID2 /\
    U := ((tpM(M1) m* sampleR'(SAM2)) v+ sampleE1(SAM2)) /\
    V := (((tpV((M1 m* V1) v+ V2) dot sampleR'(SAM2))
            v+
            sampleE2(SAM2)) v+ Decompress(SAM1,1)) /\
    C1 := Compress(U,du) /\
    C2 := Compress(V,dv) /\
    C := ([C1,C2]) .
```

**Figure 31** Conditions of rule Enc in the symbolic model of Kyber.

matrix A, obtained through function generateA, multiplied by the secret key s plus a sampled error e. For the secret key SK, we assumed it to be the output value from a CBD using the function sampleS, so no further operations are needed for its computation.

*Enc.* As in the KeyGen rule, the conditions of Enc are used to construct the needed cryptography elements. In this case, a ciphered text $c$ consisting of a pair of ciphered texts, denoted as $c_1$ and $c_2$. Both elements are specified following the operations in Fig. 14. The conditions constructing the two elements of C are available in Fig. 31. Before them, we also had to define variables U and V that represent the vectors **u** and **v**, respectively. Constant values du and dv symbolically represent the corresponding constants from the parameters of the protocol.

*Dec.* Figure 32 shows the multiple conditions necessary to compute the shared key, denoted with variable K1. The matching equations for U' and V' allow us to extract the original U and V, presented in Fig. 31, thanks to the equational theory regarding functions Compress and Decompress. Furthermore, the definition of the shared key K1 matches the equational theory for error cancellation represented by Eq. (2).

## BIKE specification

We specify the necessary equations in the functional module BIKE-DEC on the equational theory for this code-based KEM. Two main properties must be addressed to treat the ciphertext; both were explained in 'BIKE'. The first property addresses the failure rate of the decoder being zero, thus returning the original errors. The second property relates to the exclusive or operation performed over the message $m$ with the hash function $L$ over the errors.

For the specification of the always correct application of the decoder, we specify the equation

```
if  ID1 =/= ID2 /\
    ID2 =/= none /\
    U' := Decompress(first(C),du) /\
    V' := Decompress(second(C),dv) /\
    K1 := Compress(V' v- tpV(SK) dot U', 1) .
```

**Figure 32** **Conditions of rule Dec in the symbolic model of Kyber.**

```
eq decoder((P0 p* P1) p+ (P2 p* P3), P1, P3) = [P0,P2].
```
where P0, P1, P2 and P3 are variables to represent polynomials. These variables play an important role, given their positions. Notice that $c_0 = e_0 + e_1 h$, where $h = h_1 h_0^{-1} . c_0$ is the first component of the ciphertext $c$, as we saw in 'Security Fundamentals' for BIKE. Notice that decoder receives three inputs: $c_0$ multiplied by $h_0$, $h_0$ and $h_1$. If we expand and apply the distributive operation of the product over addition, we get that $c_0 h_0 = (e_0 + e_1 h)h_0 = (e_0 + e_1(h_1 h_0^{-1}))h_0 = e_0 h_0 + (e_1(h_1 h_0^{-1}))h_0 = e_0 h_0 + e_1 h_1$. This final term matches perfectly with the left-hand side of the equation we just defined. This is only possible if we know the exact values for $h_0$ and $h_1$, which are not derivable from $h$. In detail, variables P0 and P2 match with errors $e_0$ and $e_1$ respectively, meanwhile variables P1 and P3 match with $h_0$ and $h_1$ respectively.

The properties of the exclusive or operation are represented with the equations
```
eq D xorD 0 = D.
```
```
eq D xorD D=0.
```
where D is a variable to represent any of our previously defined data types. Thanks to this property, we can extract from $c_1 = m \oplus L(e_0, e_1)$ the original message $m$. This is seen in Fig. 15, specifically in line three where $c_1 \oplus L(e') = c_1 \oplus L(e_0, e_1) = (m \oplus L(e_0, e_1)) \oplus L(e_0, e_1)$, and with the application of both equations we get $m$ since the variable D represents something of type *Data*, and it is first instantiated as $L(e_0, e_1)$ with the second equation, and latter as $m$ with the first equation.

Also, new equations might arise regarding possible properties from possible BIKE implementations of the cryptographic primitives. A clear example is the encapsulation algorithm for treating the input values of the hash function L. As *Aragon et al. (2017)* dictates for this case, the procedure is first to process L(e0) and then L(e1), concatenating both results. The main issue would be that an implementer might approach this by adding the values and performing L, *i.e.,* L(e0+e1). This leads to a new equality where L(e0,e1) = L(e0+e1). This property is represented in our specification as
```
eq L([P0,P1]) = L(P0 p+ P1).
```
This equation states that when the hash function L receives a pair of values P0 and P1, then it is equivalent to applying L to the addition of P0 and P1.

*KeyGen.* On the specific conditions related to the conditional rule KeyGen, we can notice in Fig. 33 that we only symbolically represent the computation of the public key. The reason to omit the representation of any possible secret key is that it can be recovered later in terms of the components used to compute the public key. Remember, from Fig. 15 that

```
if PK := (second(SAM2) p* inv(first(SAM2))) .
```

**Figure 33** Conditions of rule KeyGen in the symbolic model of BIKE.

```
if  ID1 =/= ID2 /\
    Es := H(SAM1) /\
    E0 := first(Es) /\
    E1 := second(Es) /\
    Cs := [(E0 p+ (E1 p* (PK))), (SAM1 xorD L(Es))] /\
    SK := K(SAM1, Cs) .
```

**Figure 34** Conditions of Enc in the symbolic model of BIKE.

the KeyGen algorithm is defined to first sample two values $(h_0, h_1) \leftarrow \mathcal{DH}_w$, and uses them in the computation of the public key $h = h_1 h_0^{-1}$. Finally, it samples a value $\sigma \overset{\$}{\leftarrow} \mathcal{M}$ which is stored alongside values $(h_0, h_1)$ as a private key. We decided not to model the last sampling since it is for later use in the Dec step if the decoder fails. Since such a path is not possible because we assume zero decoding failure rate, as stated in the assumptions of 'BIKE', we simplify the model.

*Enc.* The conditions of Enc are built to model all the steps of the encapsulation without exception. Figure 34 shows, between others, the computation of the errors as E0 and E1, the definition of the ciphertext as a tuple stored in Cs and the computation of the shared key through the hashing function K using the sampled value SAM1 and the ciphertext Cs.

*Dec.* With a similar structure, the conditions of the rewrite rule for Dec are defined to represent all aspects present in the specification. Figure 35 depicts such a representation. It is important to note that since we didn't store the full secret key, we can, in a very elegant way, prepare the decoder to extract the errors from the first component of the ciphertext, as we explained before in this section with the property related to the decoder. Furthermore, the equational theory regarding the exclusive or operation helps to obtain the value m. It is now that the equational theory plays an important role, both for the decoder and for the exclusive or operation xorD.

## Classic McEliece specification

Module CM-DEC contains the equational theory representing the property between operations ENCODE and DECODE. This property was first given in Eq. (3), and the translation into Maude syntax is

```
eq DECODE(ENCODE(E,T),T') = E .
```

In this equation, we can see the use of three variables to represent different values. Variable E represents the error encoded in the encapsulation step. Variable T represents the public key used to encode the error. Finally, variable T', *i.e.,* gamma prime, represents the value used in the decapsulation step as a secret key in order to decode the encoded error

```
if  ID1 =/= ID2 /\
    ID2 =/= none /\
    Es := decoder(first(Cs) p* first(Hs), first(Hs), second(Hs)) /\
    E0 := first(Es) /\
    E1 := second(Es) /\
    M := second(Cs) xorD L(Es) /\
    SK := K(M, Cs) .
```

**Figure 35** Conditions of rule `Dec` in the symbolic model of BIKE.

```
if  E := G(SAM1) /\
    D' := lastL(E) /\
    S := firstN(E) /\
    ALPHAQ := FIELDORDERING(E) /\
    GAMMA := [IRREDUCIBLE(E), segmentN(ALPHAQ)] /\
    AUX := MATGEN(GAMMA) /\
    GAMMA' := [elem(3, AUX), elem(4, AUX)] /\
    PK := head(AUX) /\
    ALPHA := (second(GAMMA') elemN(ALPHAQ) ALPHAQ) /\
    SK := (SAM1 elem(2, AUX) first(GAMMA') ALPHA S) .
```

**Figure 36** Conditions of rule `KeyGen` in the symbolic model of Classic McEliece.

properly. This equation might seem very general and wide since multiple values with no relations between them can match with its right-hand side.

*KeyGen.* The adaptation from the specification code provided by *Chou et al. (2022)* to our symbolic model regarding the key generation step represented by rule *KeyGen* is done directly over function SeededKeyGen from Fig. 16. This function is essentially the key generation algorithm. It comprises many other functions we use to obtain certain values to construct the public and secret keys. For the public key PK, we can see in Fig. 36 that it is the resulting value at the head from the matrix generation function known as MATGEN. In the case of the secret key SK, it is not a specific value but a set of values. The set of values for the secret key comprises a sample SAM1 and some elements from the output of MATGEN and other elements that derive from them.

*Enc.* The encapsulation algorithm is straightforward. Figure 37 shows that we only need to model three values. First, the error is represented with variable ER through the result of the function FIXEDWEIGHT. Then we set variable C, which represents the ciphertext, as the encoding of the error with the public key. Finally, as with all KEMs, the shared key computation comes from one of the encapsulation step sides. To this end, we modelled the hash function H with a preset value one along the error and ciphertext as subsequent parameters.

*Dec.* The algorithm for the decapsulation is portrayed in Fig. 38. Two elements from the secret key SK are extracted to construct what is then stored in variable GAMMA'. This variable represents $\Gamma'$ from line 2 at step Dec in Fig. 16. With this newly constructed value

```
if  ID1 =/= ID2 /\
    ER := FIXEDWEIGHT(PK) /\
    C := ENCODE(ER, PK) /\
    K1 := H(1, ER, C)  .
```

**Figure 37  Conditions of rule Enc in the symbolic model of Classic McEliece.**

```
if  ID1 =/= ID2 /\
    ID2 =/= none /\
    S := last(SK) /\
    GAMMA' := [elem(3, SK), elem(4, SK)] /\
    ER := DECODE(C,GAMMA') /\
    K1 := H(1, ER, C)  .
```

**Figure 38  Conditions of rule Dec in the symbolic model of Classic McEliece.**

and the application of DECODE over C with GAMMA', we get the original error thanks to our equational theory. Finally, similar to the previous step, the shared key is computed with the same inputs, thus obtaining the same key.

*Dec.* There are some capabilities an intruder might want to perform in BIKE. These capabilities take advantage of possible design vulnerabilities, allowing an intruder to subtract certain information from another participant that might later come as valuable knowledge. The new capabilities imply two rules depicted in Fig. 39. The first rule implies that an intruder might modify or send a message where the public key sent to Bob is equal to one. This public key was not generated by the intruder using the key generation function and thus has certain implications. One implication is that upon its construction, $h_0 == h_1$, so when computing $pk = h_1 * (h_0)^{-1} = 1$. Also, with this public key, the encapsulation function becomes the identity function. The second rule represents the capabilities of the intruder to differentiate from a message the two components of the ciphertext, *i.e.*, $c_0$ and $c_1$. Also, the conditions of the second rule represent the intruder taking advantage of the design vulnerability regarding the implementation of hash function L to obtain the generated value $m_B$, which is the basis for the shared key, depicted in the rule as SK, computation.

## VERIFICATION

This section explains the verification tools we applied to the system specification defined in the previous section. Specifically, we verify our symbolic models by two methods. The first verification method is through reachability analysis. With it, we explore all the possible executions of our model and confirm no dangerous or illegal states are present. The second verification method is a more formal process called model checking. Using this tool, we

```
crl [Step1-IdentityAttack] :      {CONT3}
                                  < (Eve[KS1]CONT1) PS >
                                  net(MSGS msg{(Alice,Bob)[sentPK](PK)})
                                  =>
                                  {CONT3}
                                  < (Eve[publicKey(Alice,PK) ; KS1]CONT1) PS >
                                  net(MSGS msg{(Alice,Bob)[sentPK](1)})
                                  if PK =/= 1 .

crl [Step2-IdentityAttack] :      {CONT3}
                                  < (Eve[KS1]CONT1) PS >
                                  net(MSGS msg{(Bob,Alice)[sentC](Cs)})
                                  =>
                                  {CONT3}
                                  < (Eve[sharedKey(Bob, SK) ; KS1]CONT1) PS >
                                  net(MSGS msg{(Bob,Alice)[receivedC](Cs)})
                                  if  M := (second(Cs)) xorD L(first(Cs)) /\
                                      SK := K(M,Cs)  .
```

**Figure 39  New capabilities an intruder might use to learn certain values in BIKE without following the scheme.**

```
eq initX =  {SAMPLESX}
            < (Alice[emptyK]peer(none))
              (Eve[emptyK]peer(none))
              (Bob[emptyK]peer(none)) >
            net(emptyM) .
```

**Figure 40  Definition template of an initial state for any of our system modules representing a KEM.**

specify some properties in linear temporal logic (LTL) and use the built-in model checker in Maude to verify the symbolic models.

## Reachability verification

Using the search command, we verify if the model behaves as expected, which means checking if states of interest exist. We conduct reachability analysis from two initial states, init1 and init2. An example of the first initial state was already explained with Fig. 21 for the case of KYBER. Nevertheless, Figure 40 shows a template to define initial states in any of our specified KEMs. The second initial state, init2, defines our global state with a set of samples SAMPLEX such that each sample set has two sample values available. The extension of sample values is the main difference with the first initial state init1. The possibility of having two sample options for each sampling set allows our model to simulate two key exchange sessions of any of the KEMs. Furthermore, common to all KEMs, initX also specifies three participants populating the network, Alice, Eve and Bob, each with an empty set of keys and without a set peer. We define the network of messages as being initially empty.

For each of these two initial states, we check two things:

```
search  initX
        =>!
        { CONT4:Content }
        < (Alice[sharedKey(Bob,K1:SKey) ; KS1]CONT1)
          (Bob[sharedKey(Alice,K1:SKey) ; KS2]CONT2) PS >
        net(MSGS) .
```

**Figure 41** **Command template for a key agreement session of any KEMs of our framework in Maude.**

- Correctness of our model, *i.e.,* the existence of a state where two participants have successfully shared a key. This is achieved with the command presented at Fig. 41, where `initX` is one of the initial states. The right-hand side shows there are two **honest** participants, *i.e.,* `Alice` and `Bob`, who have succeeded in the application of the protocol, *i.e.,* they share the same key in their respective pool of keys. Such keys can only be obtained through the rewriting computation of our symbolic model thanks to the rules we defined to model the participant behaviour.
- Presence of vulnerabilities or attacks. Specifically, we search for an instance in the state space tree in which a man-in-the-middle attack has happened. This endeavour is achieved with the command depicted in Fig. 42, where the final state specifies three participants in the global state. The command can be translated so the state we are looking for has two participants with identifiers `ID1` and `ID2` that share the same key but with different participant identifiers. Moreover, a key has been shared between participants `ID2` and `ID3` in the same way, where the participant identifiers differ between them. With it, we are trying to search for states where participants `ID1` and `ID3` think they have shared the same key when in reality, both are different, thus resulting in a man-in-the-middle attack.

### Correctness

The search for states where the protocol terminates is successful for both initial states in any of the three symbolic models. This means honest participants, `Alice` and `Bob`, share the same key after following the specified rules. This demonstrates that our model works regarding the respective specifications of Kyber (*Avanzi et al., 2019*), Bike (*Aragon et al., 2017*) and Classic McEliece (*Chou et al., 2022*).

### Man-in-the-middle

Regarding the second search, it returns solutions only for the case of the second initial state `init2` over the three symbolic models. This implies some trace of state transitions, also known in the security area as an attack vector, are present during the simulations, leading to insecure states. We analyze the first solution because all other solutions fit this view. The attack vector is depicted in Fig. 43 as a pseudo-network diagram. Here, the participants in our model are displayed, with the steps of the protocol represented as boxes and the messages as directed arrows. The black dots in the malicious participant, known as EVE, are the intercept capabilities at work. As we see depicted by the diagram, when this malicious agent Eve modifies the messages sent by `Alice`, it supplants the identity since the public

```
search   initX
         =>*
         { CONT }
         < (ID1[sharedKey(ID3,K1) ; KS1]CONT1)
           (ID2[sharedKey(ID1,K1) ;  sharedKey(ID3,K2) ; KS2]CONT2)
           (ID3[sharedKey(ID1,K2) ; KS3]CONT3) >
         net(MSGS) .
```

**Figure 42** **Command template to search for a man-in-the-middle attack of any KEMs of our framework in Maude.**

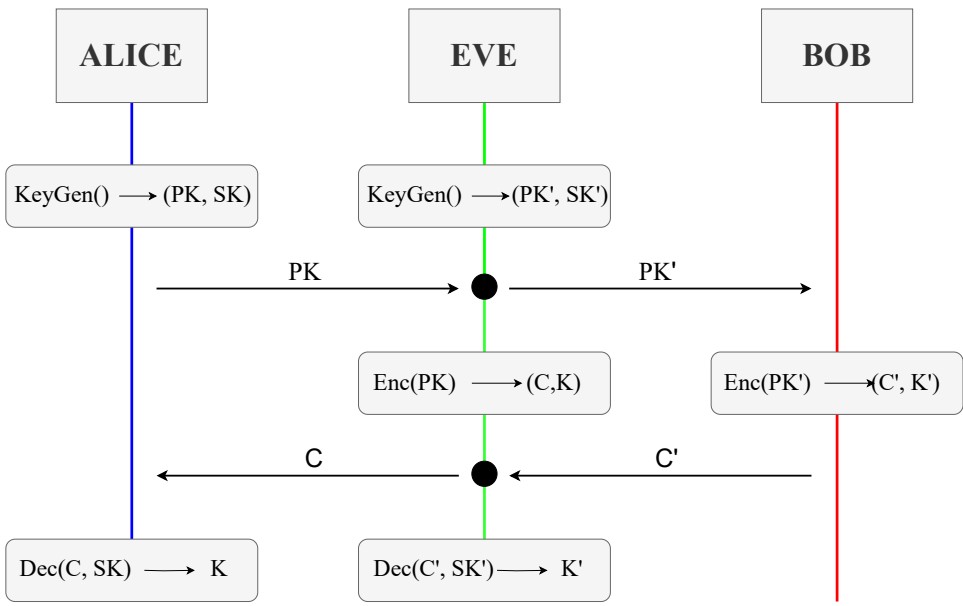

**Figure 43** **Diagram depicting a possible step trace that led to the MITM attack of a KEM regarding our symbolic model.**

key of the new message is from `Eve`. This also takes place the other way around, when the message sent by `Bob` is intercepted, and the ciphertext is modified to the one generated by `Eve` using `Alice`'s public key. In both cases, `Eve` uses the gleaned contents to its advantage. First, the gleaned public key is used in the `Enc` step to generate a shared key without `Alice` being aware. Second, after intercepting the second message, `Eve` can perform the `Dec` step with the gleaned ciphertext since it was encapsulated using her own public key. In this way, `Eve` possesses the necessary information in certain moments to carry out a man-in-the-middle attack between `Alice` and `Bob`, the two honest participants of the network.

### Design vulnerability

Some of the states that the second search command finds for the symbolic model of BIKE are due to a vulnerability in the KEM's design. As stated in 'BIKE Specification', some

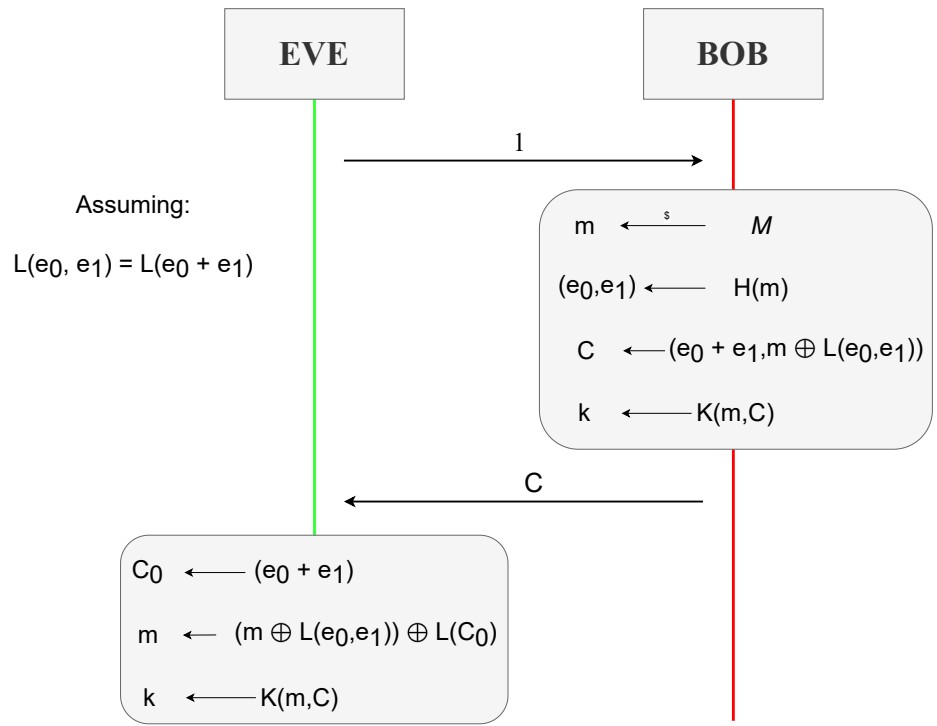

**Figure 44** Diagram depicting a step trace in the BIKE symbolic model leading to the leakage of critical information.

properties might arise due to poor implementation decisions. The combination of these new properties with an intruder taking advantage of the possible design flaws leads to cases where the intruder can learn sampling values that can later be reused due to implementation decisions.

A trace of the vulnerability can be found in Fig. 44. Here an intruder, let us refer to it as Eve, might choose to start a communication with some honest participant, Bob. Since the intruder could not follow the cryptography primitive KeyGen, Eve decides to use a public key $h = h_1 * (h_0)^{-1}$ such that $h_0 = h_1$ so $h = 1$. Once Bob receives the public key, thinking it is from Alice, he applies the encapsulation function with it and sends back a message with the resulting ciphertext. Eve then intercepts the message, and if Bob's implementation satisfies the property $L(e_0, e_1) = L(e_0 + e_1)$, then she is able to compute the shared key without the need for a decapsulation function. Eve was able to learn critical information using a weak key and a vulnerability of the exclusive or function. The specific application of the exclusive or operation can be seen at the end of the timeline of Eve in Fig. 44. Basically, Eve extracts $c_0$ from the sent ciphertext by Bob and uses it in the hash function $L$ to apply the exclusive or operation over the second part of the sent ciphertext, $c_1$.

Although this design vulnerability may not seem very relevant, it requires some extra checks by future implementations to ensure that (i) weak keys are not accepted and (ii) a proper implementation must not satisfy the $L(e_0, e_1) = L(e_0 + e_1)$ algebraic property. An extra possible solution is the use of authentication or integrity checks on the messages in

the network. This makes Bob avoid any message that is not from a trusted participant or that has been altered.

## Formal verification

The requirements for using model checking in Maude are two. The first requirement is to have some predicates specified over our model. The second requirement is to define some formulas with them. With these two fulfilled, we can then use the model-checking analysis provided by Maude. It is important to mention that we must also declare some initial states upon which the formulas would be applied. We used the same initial states previously defined using the template in Fig. 40.

### Predicates

In order to use model checking in Maude, one needs two things: a system module representing the system to check and some predicates to define properties through formulas. The system module has already been defined and tested, and now we dive into the three specified predicates for model checking.

Predicate `wantsToShareKey`, depicted in Fig. 45, is defined so it holds in a global state where a participant with identifier ID1 has his own public key, meaning he has performed the KeyGen step, and there is a message to another participant, with identifier ID2, different than him. This predicate represents a participant wanting to share a key with another. In other words, it is the start of any of the specified KEMs.

Predicate `sharedAKeyWith`, depicted in Fig. 46, is defined so it is true when two participants, ID1 and ID2, hold the same shared key K1. This predicate would then represent the end of the KEM execution, fulfilling the previous predicate in which the KEM was started.

Finally, predicate `stolenSharedKey` is depicted in Fig. 47. The predicate holds in any state where the shared key a participant has with another one is present in the pool of keys of a different participant. It is important to notice that we do not require that the third participant has or does not have the shared key.

### Properties

With the predicates defined, we now specify three properties. One security property stating something bad never happens. One liveness property to state something good eventually happens. One property to check the fairness of our system. We write these properties as LTL formulas, allowing us to explore the execution tree in search of counterexamples. If no counterexample is found, we can say with assurance that the property holds in our symbolic model.

*SECRECY.* The property concerns the assurance that the predicate of `stolenSharedKey` is false in any future state. In other words, no participant learns the secret key of another one in any state of the session. The property is specified for the cases where the secret key is from `Alice` or `Bob`, and the thief is `Eve`. The property in LTL can be written as

$(\Box \neg P)$

where $P = $ `stolenSharedKey(Alice, Eve)`.

```
eq { CONT }
    < (ID1[publicKey(ID1,PK) ; KS1]CONT1)
      (ID2[KS2]CONT2) PS >
    net(MSGS msg{(ID1,ID2)[sentPK]PK})
    |=
    wantsToShareKey(ID1,ID2) = true .
```

**Figure 45** **wantsToShareKey** predicate definition in module **KEM-PREDS**.

```
eq { CONT }
    < (ID1[sharedKey(ID2,K1) ; KS1]CONT1)
      (ID2[sharedKey(ID1,K1) ; KS2]CONT2) PS >
    net(MSGS)
    |=
    sharedAKeyWith(ID1,ID2) = true .
```

**Figure 46** **sharedAKeyWith** predicate definition in module **KEM-PREDS**.

```
eq {CONT4}
    < (ID1[sharedKey(ID3,K) ; KS1]CONT1)
      (ID2[sharedKey(ID1,K) ; KS2]CONT2)
      (ID3[KS3]CONT3) >
    net(MSGS)
    |=
    stolenSharedKey(ID1,ID2) = true .
```

**Figure 47** **stolenSharedKey** predicate definition in module **KEM-PREDS**.

*KEY SHARING.* This liveness property checks that whenever Alice wants to share a key with Bob, they eventually do so. This natural language description can be expressed in LTL notation as

$$\Box(P \rightarrow \Diamond Q)$$

where $P =$ wantsToShareKey(Alice, Bob) and $Q =$ sharedAKeyWith(Alice, Bob). Even though Eve is not specified in the formula it is present in the analyzed execution paths.

*FAIRNESS.* This property assures that whenever Alice wants to share a key with Bob, they do so infinitely many often. It is important to note that even though the malicious participant, Eve, is not explicit in the formula, it is present in the analyzed execution paths. The property in LTL can be expressed from the natural language description as

$$\Box\Diamond(P \rightarrow \Diamond Q)$$

where $P$ and $Q$ are defined as in the previous formula.

*Results*

About the execution of our LTL formulas, we have applied the three of them over our two initial states, `init1` and `init2`, for the three protocols. The results are that both initial states accomplish the liveness (KEY SHARING) and fairness property when `Alice` and `Bob` want to share a key, in any of the three KEMs symbolic models. We get different results per the security property concerning SECRECY. For the KEM's symbolic models of Classic McEliece and Kyber, the property holds in the initial state `init1`. In the case of the initial state `init2`, it does not hold due to the MITM attack that was reported with the reachability analysis. In the case of the symbolic model of BIKE, the only case where the property holds is in the initial state `init1` when the participant is `Alice` and the thief is `Eve`. The property does not hold when the participant is Bob, due to the design's vulnerability, and when the initial state is `init2`, because of the MITM attack.

To replicate these experiments, in the GitHub repository, there is a text file named as each of the KEMs. These documents present the commands one needs to execute to replicate the search and model checking we have presented.

# CONCLUSION

This article provides a framework for the symbolic specification and analysis of lattice and code-based KEMs. We use our framework to symbolically specify and analyze three KEMs, one being lattice-based and the other two being code-based. We have proven the presence of a man-in-the-middle attack on the three KEMs through reachability analysis in Maude. Furthermore, to extend our model's verification, three LTL formulas, specifying liveness, fairness and security properties, have been applied with Maude's LTL Model Checker. With the verification results, we can assure that Kyber, BIKE and McEliece are not safe from classical adversaries if no authentication or integrity of the messages is available or defined. Furthermore, BIKE suffers from a design vulnerability that should not be left unchecked. We propose, as a solution, the inclusion of some form of check over the encapsulation function to avoid using insecure or weak keys.

For future work, we plan to improve the analysis by conducting extended model checking to verify the complete correctness of our symbolic models. We also aim to extend the framework in order to represent the key encapsulation mechanism properties better. We also consider using protocol analysis tools, such as Maude-NPA, to specify these or other KEMs and check their security in a more thoughtful analysis, *i.e.,* for an unbounded number of sessions. We could also extend the system representation to use the objects feature from Maude, making it closer to other high-level languages and more understandable for non-experts in formal methods. Using this new feature, we could specify multiple layers of protocols and check the interaction between them. For example, add capabilities of authentication or signatures above any KEM and perform the analyses we have made, in this article, to check if the results are equivalent.

### Funding

Víctor García and Santiago Escobar were supported by the grant PID2021-122830OB-C42 funded by MCIN/AEI/10.13039/501100011033 and ERDF A way of making Europe and by the grant PCI2020-120708-2 funded by MICIN/AEI/10.13039/501100011033 and by the European Union NextGenerationEU/PRTR. Kazuhiro Ogata was supported by JST SICORP Grant Number JPMJSC20C2, Japan. Sedat Akleylek was supported by TUBITAK under Grant No. 121R006. Ayoub Otmani was supported by FAVPQC project funded by CNRS and by the grant ANR-22-PETQ-0008 PQ-TLS funded by Agence Nationale de la Recherche (ANR) within France 2030 program. There was no additional external funding received for this study. The funders had no role in study design, data collection and analysis, decision to publish, or preparation of the manuscript.

### Grant Disclosures

The following grant information was disclosed by the authors:
MCIN/AEI/10.13039/501100011033: PID2021-122830OB-C42.
ERDF A way of making Europe.
MICIN/AEI/10.13039/501100011033: PCI2020-120708-2.
European Union NextGenerationEU/PRTR.
JST SICORP: JPMJSC20C2, Japan.
TUBITAK: 121R006.
FAVPQC project funded by CNRS: ANR-22-PETQ-0008 PQ-TLS.
Agence Nationale de la Recherche (ANR) within France 2030 program.

### Competing Interests

Sedat Akleylek is an Academic Editor for PeerJ.

### Author Contributions

- Víctor García conceived and designed the experiments, performed the experiments, analyzed the data, performed the computation work, prepared figures and/or tables, authored or reviewed drafts of the article, and approved the final draft.
- Santiago Escobar conceived and designed the experiments, analyzed the data, authored or reviewed drafts of the article, and approved the final draft.
- Kazuhiro Ogata conceived and designed the experiments, analyzed the data, authored or reviewed drafts of the article, and approved the final draft.
- Sedat Akleylek analyzed the data, authored or reviewed drafts of the article, and approved the final draft.
- Ayoub Otmani analyzed the data, authored or reviewed drafts of the article, and approved the final draft.

### Data Availability

The source code is available at GitHub and Zenodo:

- https://github.com/v1ct0r-byte/PQC-in-Maude/releases/tag/v1.0.0.
- Víctor García Valero. (2023). v1ct0r-byte/PQC-in-Maude: PeerJ Computer Science (v1.0.0). Zenodo. https://doi.org/10.5281/zenodo.8099002.

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
