# Peer review of "Modelling and verification of post-quantum key encapsulation mechanisms using Maude"

_PeerJ Computer Science, doi:10.7717/peerj-cs.1547_

## Round 0.1 · original submission · Major Revisions

Experts have now judged your manuscript. You are asked to undertake a major revision in which you carefully take all comments and suggestions of the reviewers into account and answer them in a response letter. In particular, please carefully state the paper's main contributions, making it crystal clear why the reported research is relevant and meaningful and what exactly constitutes the increment with respect to earlier work by some of the authors (most specifically Tran et al.).

Reviewer 1 ·

Basic reporting

Summary:

This paper is on modelling and verification of post-quantum key encapsulation mechanisms using Maude. The authors consider three key encapsulation mechanisms which have been selected as post-quantum security schemes. These mechanisms are translated into a model and implemented in the model checker Maude. This analysis shows the presence of a man-in-the-middle attack on them and other vulnerabilities of these key encapsulation mechanisms.

First of all, this paper is well-written in clear English language. This is an extended version of the workshop paper of García et al (2022) "Modeling and verification of the post-quantum key encapsulation mechanism KYBER using Maude" and the master thesis of the first author with the same title. Moreover, in my humble opinion, this paper seems to be incremental with respect to Tran et al. (2022b/c).

This paper may not entirely fall within the scope of PeerJ Computer Science as it is a case study in cryptoanalysis, only using some tools from computer science, like model checking.

From the paper, it is not clear why this research is relevant and meaningful. Specifically, it's not clear what are the concrete implications of the results, as the authors found vulnerabilities to the investigated mechanisms. For instance in Section 7:
"Furthermore, BIKE might suffer from a design vulnerability that should not be left unchecked."
Why "might?" What consequences can this have? This should be explained further.

It is not clear from the introduction and the related work section what symbolic analysis adds to computational analysis. The authors should also explain more thoroughly why symbolic analysis is useful for security checks of cryptoschemes e.g. with respect to computational analysis, as this is what the add compared to Tran et al. (2022b/c).

The Maude implementation code explanation and the description of the experimental method are intertwined e.g. in section 6 and 7.1, which made it hard for me to read the experimental parts.
The authors should consider separating these subjects, or even moving the explanation of any Maude code to an appendix.



Typos and remarks:

Line 33: NP stands for non-deterministic polynomial time. It's not known whether problems in NP can be solved in (non-)polynomial time, as this is the famous P=NP problem.

Line 40: It's better to cite the original Grover paper from 90s instead of Grassl et al. as the quadratic complexity reduction of the integer factorization problem follows almost directly from this.

Line 87: Currently,

Figures 2 - 11, 17 - 29: Consider to put these figures inline as this can improve readibility.

Figure 10: You can add add line numbers to the code. This makes it easier to follow as on line 363 you refer to specific lines.

Line 439: Shouldn't this be delta < 2^{-139}? It is not clear why it's relevant to state the decryption failure probability in this article.

Line 444: See comment on line 33. It's not clear to me why an NP-problem is unsolvable by quantum computers and I am not sure whether this is actually true. Please provide a reference.

Lines 711 and 713 and 743 and 751: remove starting comma.

Line 927: It is not to me clear what is meant by the initial states initial1 and initial2. Please explain.

Line 1052 - 1057: One article is cited twice in two different versions. Please merge them into one version.

Experimental design

This is more of a case study.
Computational security and symbolic security are checked using maude.

Validity of the findings

The maude code is available as open source on a web page which increases confidence in the validity.
We did not try to reproduce the findings.
https://github.com/v1ct0r-byte/PQC-in-Maude

Reviewer 2 ·

Basic reporting

The paper presents a Maude framework to model and verify protocols for key encapsulation and applies it to the verification of three post-quantum mechanisms which participated in the recent NIST competition.

The paper is well-written overall, and the research presented in this paper is quite interesting and timely. I suggest the paper undergoes a revision because some small issues need to be addressed before final acceptance.

In particular, my main concerns are:
1. Although the paper is well written, some sentences must be revised.
- Line 22: in the sentence “behaviour of each of them”, it is unclear to which the pronoun “them” is referring. I assume you mean “the participants of the protocol”, but it is better to clarify it.
- Lines 49 - 60: the explanation is not linear. I would first explain the mechanism of key encapsulation, then say about the three proposals, and then talk about the attacks.
- Line 202: “Each chapter” should be “Each section”.
- Line 421: the sentence “we … so we can” is unclear. I did not understand the message you wanted to convey. Please, rephrase it.
- Line 459: the sentence “showing … we explained” is unclear. In particular, it is unclear what “showing” refers to.
- Line 491: “A lecturer with an open eye” is unclear. Do you mean “A careful reader”?
2. The introduction sets the context appropriately, but I would expect one or more sentences that explicitly say what the main contributions/findings of the paper are. For example, you can add a sentence just before the plan of the paper, such as “In summary the main contributions of the paper is/are” and then a statement or a list of bullet points with the state of the contributions.
3. Section 3 contains the background, however, also Section 4 contains information about the protocols considered, so it is also background. My suggestion is to include Section 4 as a subsection of Section 3.
4. In Section 3, some snippets of the code appear inline in the text, such as the one in lines 218 and 220. They are not highlighted well, so I suggest putting them in a dedicated line.
5. The majority of figures are fine, I have some concerns for a few of them. In particular, Figure 13 is a bit overcrowded and requires some explanation in the text: I would explain what Algorithm 7 does and which participant runs it. Then I would explain that its result is the public key sent to the other participant, which in turn uses it to run Algorithm 8. The same is true for Algorithm 9. In Figure 14, there are no arrows, so it is not clear what data are exchanged between the two participants, so that I would add the missing arrows together with an explanation in the text. Figure 15 and Figure 16 use different notations for presenting the algorithms, so I would make them uniform with the previous figures.

Other minor comments:
1. In Section 5.2, when you present the verification framework, you could add a picture describing its architecture. This could make the rest of the section clearer.
2. Section 6 has some spurious commas and dots at lines 711, 751, 756, 779. I would remove them.

Experimental design

The code is available on a GitHub repository. This is good because it can allow others to reuse the framework for either reproducing the research or studying the security of other protocols. However, I would have expected a more detailed README that explains how the repository is organized and how to run the experiments described in the paper. I would also consider Zenodo to archive the code.

Validity of the findings

The results presented in the paper can be reproduced in theory because the code is available on GitHub. However, I would have expected some instructions in the repository explaining how to run the experiments described in the paper. The conclusions are well-stated and linked to the research questions.

---

## Round 0.2 · accepted · Accept

All comments from both reviewers have been dealt with in a satisfactory manner by the authors and both reviewers agree that the manuscript is now ready for publication.

Reviewer 1 ·

Basic reporting

The reviewer's comments have been processed to satisfaction.

Experimental design

The reviewer's comments have been processed to satisfaction.

Validity of the findings

The reviewer's comments have been processed to satisfaction.

Additional comments

The reviewer's comments have been processed to satisfaction.

Reviewer 2 ·

Basic reporting

The revised version of the manuscript addressed the issues raised in the previous review.

Experimental design

The revised version of the manuscript addressed the issues raised in the previous review.

Validity of the findings

The revised version of the manuscript addressed the issues raised in the previous review.